# CrystalGym: A New Benchmark for Materials Discovery Using Reinforcement Learning

## Abstract

*In silico* design and optimization of new materials primarily relies on high-accuracy atomic simulators that perform density functional theory (DFT) calculations. While recent works showcase the strong potential of machine learning to accelerate the material design process, they mostly consist of generative approaches that do not use direct DFT signals as feedback to improve training and generation mainly due to DFT's high computational cost. To aid the adoption of direct DFT signals in the materials design loop through online reinforcement learning (RL), we propose **CrystalGym**, an open-source RL environment for crystalline material discovery. Using CrystalGym, we benchmark common value- and policy-based reinforcement learning algorithms for designing various crystals conditioned on target properties. Concretely, we optimize for challenging properties like the band gap, bulk modulus, and density, which are directly calculated from DFT in the environment. While none of the algorithms we benchmark solve all CrystalGym tasks, our extensive experiments and ablations show different sample efficiencies and ease of convergence to optimality for different algorithms and environment settings. Additionally, we include a case study on the scope of fine-tuning large language models with reinforcement learning for improving DFT-based rewards. Our goal is for CrystalGym to serve as a test bed for reinforcement learning researchers and material scientists to address these real-world design problems with practical applications. We therefore introduce a novel class of challenges for *reinforcement learning methods dealing with time-consuming reward signals*, paving the way for future interdisciplinary research for machine learning motivated by real-world applications.

## 1 Introduction

Reinforcement learning (RL) methods have demonstrated immense success for complex decision-making problems, robotics (Khan et al., 2020; Xu et al., 2024), autonomous driving systems, and language models (Liu et al., 2023). Recently, the scope of RL has expanded to a variety of scientific areas including energy optimization, quantum systems (Martín-Guerrero & Lamata, 2021), scientific discovery (Vinuesa et al., 2024), biology, and neuroscience. RL applications in chemistry have been studied for tasks such as molecular design, geometry optimization, and retrosynthesis (Sridharan et al., 2024). Yet, RL has been investigated on such applications only on a limited scale because of four main reasons. First, the diversity of the chemical applications means there is no standardized way of formulating the problem from an RL perspective. Every practitioner models their specific problem differently, and proposes solutions tailored towards said problem. It is thus hard to assess if results from one problem apply to another one. Second, next to the required RL expertise, domain expertise is also required to benchmark and evaluate performances on chemistry domains. Both these reasons result in a high barrier of entry for investigating RL-based methods on chemistry applications. Third, the synthetic nature of formulating chemistry applications as sequential decision-making problems, and the immensely diversified nature of the chemical space produce policies that are hard for humans to understand and interpret, especially compared to games and robotics. Fourth, these chemical applications offer unique challenges that have been less studied in the RL literature, such as noisy and time-consuming reward-signals.

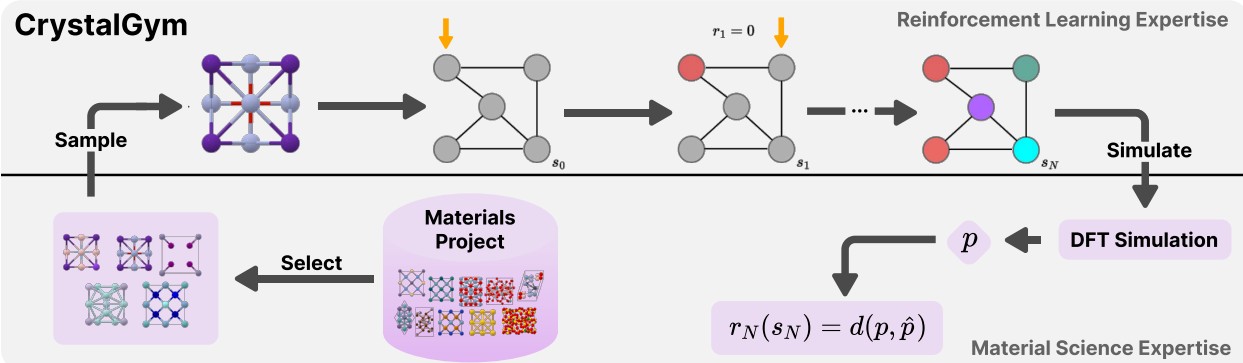

Figure 1: The **CrystalGym** environment. We select simple cubic crystals from the Materials Project (Jain et al., 2013) database. An episode starts by sampling a crystal structure from this selection. At each step, the agent selects an atom to fill a specified position. The episode ends once all positions are filled, at which point the crystal is evaluated with DFT. The parameters of the simulator are pre-set, such that they converge in reasonable time for a wide range of compositions. The reward function is computed based on a distance metric given a target value.

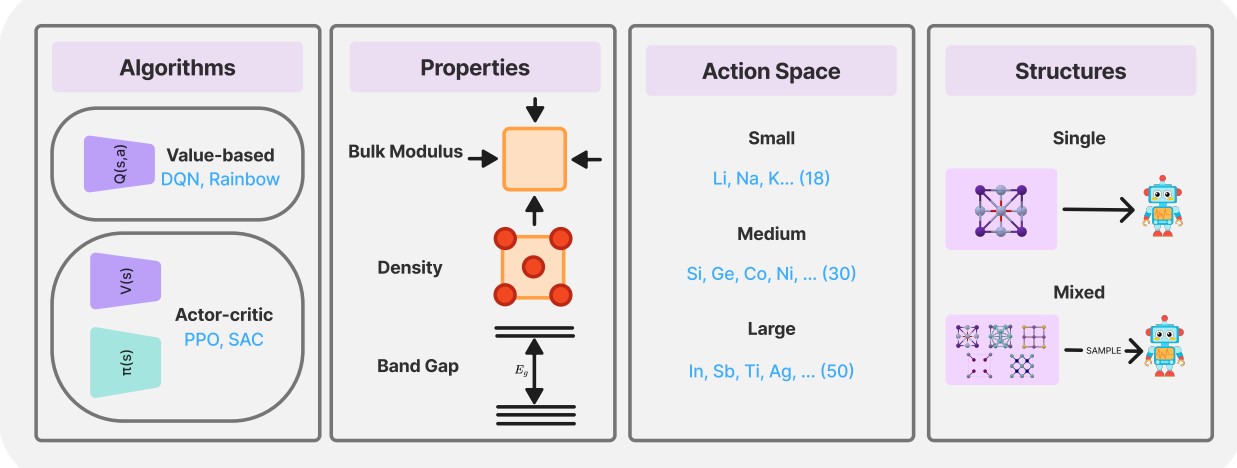

Figure 2: We evaluated four RL algorithms: PPO, DQN, Rainbow, and SAC, for optimizing material properties such as bulk modulus, density, and band gap. The action space consists of different subsets of the periodic table, with increasing levels of complexity. Additionally, the environment supports two initialization modes: starting from the same crystal lattice in every episode or sampling a lattice from a set of candidate lattices.

Material discovery is one of the applications affected by these challenges. Accelerating material discovery is an important avenue within scientific discovery, the applications of which include designing sustainable and industrially useful materials (Miret et al., 2024). This often involves optimizing for a set of desired properties, computed using physical simulators based on first principles. Density Functional Theory (DFT) (Jones, 2015) is a modeling method that simulates atomic-level properties of molecules and materials using quantum mechanical laws. Considering the time-consuming nature of DFT calculations and the expertise required to operate them, most of the existing generative and language models for material generation do not incorporate them as feedback for material optimization (Gruver et al., 2024; Ding et al., 2025; Levy et al., 2025; AI4Science et al., 2023). However, the predictions made by ML models are very different from DFT computations. For example, the generative models from Gruver et al. (2024), **?** produce crystals of which 50% are stable according to M3GNet, the state-of-the-art predictor for stability, while only 11% are

actually stable according to DFT calculations. Reinforcement learning offers a complementary approach to this problem by directly learning from signals obtained from DFT, potentially without relying on large datasets. Given the dearth of dedicated RL environments and benchmarks for material discovery problems and the domain expertise involved in them, it is difficult for practitioners to investigate RL approaches for these tasks. To this end, we propose **CrystalGym**, a novel and open-source RL environment for crystalline material discovery that offers a way to learn optimal policies from rewards obtained directly from DFT. We focus particularly on optimizing the composition of a crystal by framing it as a sequential decision-making problem and formulating a deterministic Markov Decision Process (MDP), as initially proposed by Govindarajan et al. (2024) – the agent sequentially places a chemical element at a given atomic site in a crystal. We design reward functions based on DFT outputs, individually targeting three challenging properties – band gap, bulk modulus, and density. As the methods and parameters for DFT calculations are preset, they need not be modified unless necessary, hence making it easier for RL practitioners to adopt the environment without explicitly focusing on the correctness of domain-related aspects.

The goal of this environment and benchmark is to design and test RL algorithms for optimizing DFT-based rewards, and to promote future research in this new class of tasks involving optimization of time-consuming reward signals. We provide tasks where the RL agent is expected to explore the exponentially large chemical search space and drive the policy toward designing high-reward crystals. Our work considerably differs from previous works on crystal generation that used generative models (Zeni et al., 2025; Levy et al., 2025; Jiao et al., 2023) without involving DFT in the training loop or active learning works that do not attempt to optimize for functional materials properties(Merchant et al., 2023). The electronic and elastic properties we optimize for have plenty of practical and industrial applications, including efficient semiconductor and battery design, photovoltaics, and hydraulic and aerospace materials. Overall, material discovery directly influences sustainability and climate change mitigation. Our unique contributions to this work are as follows.

1. Open-source RL environment for crystal discovery based on the `Gymnasium` framework (Towers et al., 2024), that is ready to be adopted and customized by the RL and material science community.

2. Extensive analysis on performance and sample efficiency with different RL algorithms including proximal policy optimization (PPO), soft actor-critic (SAC), Rainbow, and deep Q-networks (DQN) (Schulman et al., 2017; Haarnoja et al., 2018; Hessel et al., 2018) with appropriate graph networks for the policy. We also investigate the performance of RL-based LLM fine-tuning for crystal generation using our environment.

3. We highlight several domain-related challenges in applying RL for material discovery and in general, problems that involve time-consuming and noisy reward signals, leading to potentially interesting future directions.

## 2 Background

**Related work** Crystalline material generation has gained significant attention in recent years, with generative and language models being more prominent in this space. Diffusion-based models have been proposed to learn a generative distribution from a dataset of crystals. CDVAE (Xie et al., 2022) was one of the first approaches in this area, which follows an encoder-decoder model with a denoising diffusion process, generating both the structure and composition of crystals. This was followed by models that incorporate symmetric inductive biases, such as DiffCSP (Jiao et al., 2023) and SymmCD Levy et al. (2025). Matter-Gen (Zeni et al., 2025) also used a diffusion model and performed post-training optimization for properties like band gap, bulk modulus, and magnetic density. Large language models such as Crystal-LLM (Antunes et al., 2024), Crystal-Text-LLM (Gruver et al., 2024) and MatExpert (**?**) are autoregressive approaches that used text-based representation of crystals in the 3D space. While most of these approaches evaluated the generated samples with DFT, none of them optimized for properties directly computed with DFT or used it as feedback for improving learning. Further, while GNOME (Merchant et al., 2023) used an active learning approach for material generation by optimizing for stability with DFT calculations in the loop, it did not focus on other important electronic and mechanical properties. Govindarajan et al. (2024) introduced a novel framework to optimize crystal composition for properties like formation energy and band gap using a

reinforcement learning setup, where offline learning was done to mitigate the time-consuming nature of DFT calculations. In our work, we adopt their MDP framework to build an environment and test bed for online RL algorithms. Our work is also loosely related to ChemGymRL (Beeler et al., 2024), the first interactive RL environment focusing on chemical discovery based on a simulated laboratory. Evolutionary search methods have been traditionally employed for crystal structure optimization, with representative approaches including USPEX (Glass et al., 2006) and XtalOpt (Lonie & Zurek, 2011). Unlike these methods, CrystalGym frames compositional optimization as a sequential decision-making problem where an RL agent learns to select atomic species one site at a time. Incorporating evolutionary search as an alternative optimization strategy within CrystalGym is nonetheless a natural and promising future direction.

**Crystalline materials** Crystalline materials are everywhere, from the photovoltaic cells of a solar panel to the semiconductors in every chip. They are characterized by a periodic arrangement of atoms in the 3-dimensional space. They are usually described by a lattice, represented by a unit cell with vectors $\mathbf{l_1}, \mathbf{l_2}, \mathbf{l_3} \in \mathbb{R}^3$ of length $a, b, c \in \mathbb{R}$, such that for any atom $u$ at position $\mathbf{x_u}$ in the unit cell, $u$ appears again at every position $\{\mathbf{x_u} + n_1\mathbf{l_1} + n_2\mathbf{l_2} + n_3\mathbf{l_3} \,\forall\, n_1, n_2, n_3 \in \mathbb{Z}\}$ in the lattice. The lattice displays various degrees of symmetry encompassed in the space group of the crystal, ranging from 1 to 230, where higher space group means higher level of symmetry. While recent works focused on generative and language models for generating a crystal's lattice, atomic positions, and compositions together, we simplify the problem such that the agent only predicts the identities of the atoms given fixed lattice and atom positions. This also aligns with the goal of high-throughput virtual screening (HTVS) (Jain et al., 2011), where atoms are combinatorially substituted in known crystal structures and validated using DFT to design new materials computationally. Hence, we formulate the RL problem with discrete action spaces. The scope of this study is limited to cubic crystals (space groups 200-230) with 4-8 atoms, for which DFT calculations are faster and certain properties are easier to compute.

**Reinforcement learning** In *reinforcement learning (RL)*, an agent learns to optimize its behaviour by interacting with the environment. Such a setting is modeled as a *Markov decision process (MDP)*, a tuple $\mathcal{M} = \langle \mathcal{S}, \mathcal{A}, \mathcal{T}, R, \gamma \rangle$, with state space $\mathcal{S}$, action space $\mathcal{A}$, transition probabilities $\mathcal{T}(\boldsymbol{s'}|\boldsymbol{s}, \boldsymbol{a}) : \mathcal{S} \times \mathcal{S} \times \mathcal{A} \to [0, 1]$, reward function $R(\boldsymbol{s}, \boldsymbol{a}) : \mathcal{S} \times \mathcal{A} \to \mathbb{R}$, and discount factor $\gamma \in [0, 1]$. At timestep $t$, the agent is in state $s_t$, and selects action $a_t$ using a policy of the form $\pi(a_t \mid s_t)$. Under the policy $\pi$, we call $V^\pi(s) = \mathbb{E} \sum_t [\gamma^t r_t \mid \pi, s_t = s]$ the *value*, i.e., the expected sum of discounted rewards (or return). The policy that maximizes the value is said to be the optimal policy $\pi^* = \max_\pi V^\pi$. Closely related to the value is the $Q$-value, $Q^\pi(s, a) = \mathbb{E} \sum_t [\gamma^t r_t \mid \pi, s_t = s, a_t = a]$. One of the common approaches to learn $Q^*$ is through the *Bellman equation*, $Q(s_t, a_t) \leftarrow (1 - \alpha)Q(s_t, a_t) + \alpha\delta$, where $\delta = r_t + \gamma \max_a Q(s_{t+1}, a)$ is often referred to as the *temporal-difference target*. Deep Q-networks (DQN) approximate $Q$ with a neural network $Q_\theta$ parametrized by $\theta$, by minimizing $(Q_\theta(s_t, a_t) - \delta_{\theta'})^2$, where $Q_{\theta'}$ is a periodically updated copy of $Q_\theta$ used to stabilize learning (Mnih et al., 2015). Many recent value-based algorithms still use DQN as their foundation, which is why we use it throughout our experiments to compare the different settings we introduce. Notably, Rainbow (Hessel et al., 2018) integrates the improvements of multiple DQN extensions, such as Dueling DQN (Wang et al., 2016), Double DQN (Van Hasselt et al., 2016), and prioritized experience replay (Schaul et al., 2015) into a single algorithm. Next to value-based algorithms, we also evaluate alternative approaches, such as actor-critc methods, that learn a policy explicitly. Soft actor-critic (SAC) (Haarnoja et al., 2018) is an off-policy method that learns both a policy and a Q-function, optimizing for maximum entropy to encourage exploration. Proximal policy optimization (PPO) (Schulman et al., 2017), is an approximation of trust region policy optimization methods that constrain the size of the policy update – the loss function is based on a clipped surrogate objective.

## 3 The CrystalGym environment

Our goal is to encourage the use of RL for material discovery. While deep learning methods tend to generate samples similar to their training data (Levy et al., 2025; Zeni et al., 2025), RL enables broader exploration of chemical space to uncover novel structures. For materials scientists, this allows direct optimization of DFT-calculated properties without relying on often inaccurate ML proxies (Ghugare et al., 2024; Lee et al., 2023; Bihani et al., 2024; Miret et al., 2023). For RL researchers, CrystalGym presents a unique challenge:

a synthetic transition function (interacting with the environment is a virtually free operation), but a noisy, costly reward signal. Much of the required domain expertise is baked in the environment, allowing for RL researchers to focus on algorithmic improvements for scientific discovery.

## 3.1 Crystal generation as a Markov decision process

In CrystalGym, the agent optimizes the composition of a crystal structure for a desired property value. Starting from an empty structure, the agent iterates over each position and selects an atom to place. Once all positions are filled, the episode ends and the resulting crystal is evaluated with a DFT calculator. By training on a pool of different crystals, sampled randomly at the start of each episode, CrystalGym aims to provide a generalizable RL agent, that accurately fills atomic positions even on unseen crystals. The RL agent can also specialize on a single crystal structure, by always sampling it at each episode. We adopt the deterministic MDP formulation initially proposed by Govindarajan et al. (2024). We represent crystals using graphs, with atoms as nodes and edges connecting neighboring or bonded atoms. Consider a graph $\mathcal{G}(V, E)$, with nodes (atom positions) $V = \{v_0, \ldots, v_{N-1}\}$ and edges (connections to other atom positions) $E$. Each atom position $v_j$ has a label that is either empty ($a_\varnothing$) or set to an atom-type $a_i$, where $i$ is the index of the $i$-th element of the periodic table. We consider the state-space $\mathcal{S}$ the empty, partially or fully filled graphs $\mathcal{G}$, with the initial state $s_0$ crystal $\mathcal{G}_0$, where $v_j = a_\varnothing, \forall j \in \{0..N-1\}$. The action-space $\mathcal{A}$ is defined as the atomic elements $a_i$ of the periodic table. Finally, we transition from state $s_t$ to $s_{t+1}$ by setting $v_j$ to the selected atom $a_i$. The environment inherits the `Gymnasium` framework and can be easily imported for testing RL algorithms. After every episode, DFT is used to evaluate properties of interest, and then used to compute the reward, which we detail in Section 3.2. The sequence of steps and parameters that DFT calculation requires are preset for each of the properties of interest (i.e., bulk modulus, density, and band gap). Hence, the user just needs to provide the choice of property and the desired target value, without modifying the internal DFT workflow (unless necessary). The workflow of CrystalGym and its different aspects are shown in Figure 1 and Figure 2.

## 3.2 Crystal properties and rewards

We focus on individually optimizing the composition of one or more crystals for three different crystal properties – band gap, bulk modulus, and density. The aforementioned applications require the different properties to have specific target values (as opposed to being maximized, as is typically the case for RL reward functions). Thus, for each property, we design a reward function based on the magnitude and range of the values the property can have, that encourages to be closer to a target value, $\hat{p}$. We also incorporate a penalty term if the DFT computation fails due to technical or convergence issues. Table 4 summarizes the reward formulations and their ranges for each target property, along with DFT computation times and failure rates. DFT single-point SCF calculations were performed using Quantum Espresso v7.1 (Giannozzi et al., 2009) with the PBE functional and SSSP v1.3.0 pseudopotentials (Prandini et al., 2018); full computational settings are provided in Appendix B.5.

**Bulk modulus** The bulk modulus is an elastic property of a solid-state material that measures its resistance to change in volume due to bulk compression. This property is useful in many applications involving aerospace engineering and structural design. We compute the bulk modulus by performing multiple DFT simulations after introducing small volume changes to the original crystal. We then fit a Murnaghan equation of state (Murnaghan, 1944) with the obtained energy values and corresponding volumes. Since we are mostly interested in values between 100 GPa and 1000 GPa, we choose a scaled linear function based on the absolute distance of the computed value $p_{BM}$ from the target $\hat{p}_{BM}$, i.e., $r(s_N) = \max\left(-\frac{|p_{BM} - \hat{p}_{BM}|}{\hat{p}_{BM}}, -5\right)$ if DFT is successful, and $-5$ otherwise.

**Density** We calculate the volumetric mass density of a crystalline material (in $g/cm^3$), using a single-point DFT calculation followed by 1 step of structure relaxation. As per the Materials Project Database, the density values range from 0 to 28 $g/cm^3$. Hence, we use an exponential distance function, i.e., $r(s_N) = \exp\left(-\frac{(p_D - \hat{p}_D)^2}{\hat{p}_D}\right)$ if DFT is successful, and $-1$ otherwise.

**Band gap** Band gap refers to the energy gap between the valence and conduction bands in solids, with values of interest usually in the semiconductor range, i.e., 0 eV to 5 eV, given its applications in electronics. Given a desired target value $\hat{p}_{BG}$ and computed value $p_{BG}$, we choose an exponential reward formulation, i.e., $r(\boldsymbol{s_N}) = \exp(-(p_{BG} - \hat{p}_{BG})^2)$ if DFT is successful, and $-1$ otherwise.

## 4 The CrystalGym benchmark

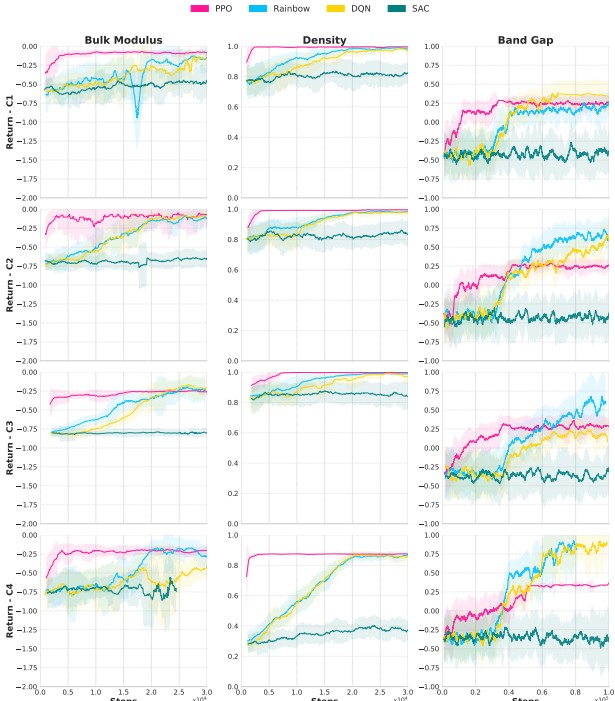

Figure 3: Learning curves for 4 of the 7 crystal structures in the simplest CrystalGym benchmark (**single** structure, *in-d.* targets, **small** action space). Task difficulty varies with both property type and crystal structure.

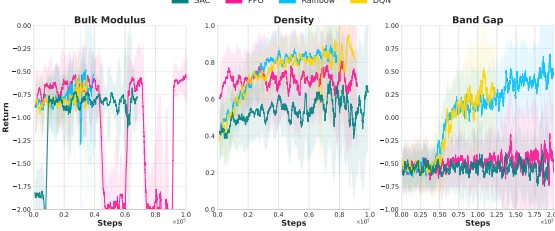

Figure 4: Training curves for experiment with **mixed** starting structures, **small** action space and *in-d.* targets.

Figure 5: Training curves for experiment with **mixed** starting structures, **small** action space and *o.o.d.* targets.

In Section 3, we have presented how we can frame crystal composition as a sequential decision process, and multiple properties of interest to optimize on. Ideally, an RL agent trained on certain crystal structures and optimized for a specific target property should be able to effectively predict the atomic identities for any relevant crystal structure, such that the resulting (filled) crystal's property value matches the target. This, however, is an extremely hard task, not only due to the diversity of the potential crystal structures and chemical space, but also due to the potentially prohibitive computation time required to execute the many DFT calculations encountered during training.

To make measurable progress on this problem, we pair our proposed CrystalGym environment with an associated benchmark. We select 7 different cubic crystals (Figure 8) from the Materials Project database (Jain et al., 2013). These crystals have a different number of atoms (4-8), and belong to 5 different space groups, which ensures the genericity of the learning algorithm. We envision multiple degrees of increasing difficulty, spread across 3 different axes of the design problem. First, agents need to be able to optimize crystals for both in-distribution and out-of-distribution property values. For the three properties of interest (bulk modulus, density and band gap), we thus specify *in-d. targets* (in-distribution) and *o.o.d. targets* (out-of-distribution). We specify their concrete values in Table 1. Second, agents should not only learn the optimal composition for the crystal structure they have been trained on, but also for novel, unseen crystals. Thus, we devise a *single structure* setting, aimed to assess the feasibility of the desired target property, and a

more complex *mixed structure* setting – where the policy is trained on 5 crystals, and evaluated on the 2 remaining ones – to measure how generalizable policies are. Third, agents should be able to freely select any atom from the periodic table, regardless of how unlikely it is to result in an optimal crystal composition. However, in practice, this results in a stark increase of failure rates of DFT calculations. To alleviate this and, consequently, speed up the learning process, we select subsets of atoms that are less prone to failure. The *small action-space* consists of 18 elements of the periodic table, primarily metals and some nonmetals of group 1 and 2, and no transition elements. The more flexible *medium action-space* contains 30 elements, and is a superset of the small action-space with additionally certain metalloids and frequently occuring transition elements, according to the Materials Project database. The full list of selected atoms is available in Appendix B.1.

We believe that, by progressing on the ***mixed structure* with *o.o.d. targets* and *small action-space* setting**, we will also make progress towards the overall goal: designing RL algorithms for material discovery, that can reliably fill any crystal structure for a desired property value. This setting, for which we share initial findings in Section 5.2, strikes a balance between the complexity of the tackled problem and the feasibility of the training procedure in terms of walltime. Notwithstanding, one could use simpler settings that focus on a specific difficulty axis while designing new RL algorithms (e.g., *single structure* with *in-d. targets* and *small action-space* in an active learning scenario designed to minimize the number of DFT calculations).

## 5    Benchmark performance and results

Having defined the CrystalGym benchmark, we now perform a set of experiments and ablations to better understand its properties and characteristics. We focus on two important aspects. First, our goal is to test the feasibility of using RL for the crystal composition completion task (Section 5.1). RL has been understudied for material generation, and DFT signals are known to be complex, so it is important to validate that they can be used as a reward signal. Second, we aim to investigate the evolution of the learning ability when the difficulty of the task is increased (Section 5.2). Following a comprehensive analysis on these variations, we finally evaluate RL-based methods on the proposed benchmark setting, providing the current state of RL for crystal generation (Section 5.2).

**Experimental setup**    For all our experiments, we compare the performance of popular value- and policy-based RL algorithms, namely proximal policy optimization (PPO) (Schulman et al., 2017), soft actor-critic (SAC) (Haarnoja et al., 2018), Rainbow (Hessel et al., 2018), and deep Q-networks (DQN) (Mnih et al., 2015) agents. Additionally, since our crystals are represented as graphs, it is convenient to adopt graph neural networks (GNN) for representation learning (Duval et al., 2023). We leverage MEGNet (Chen et al., 2019), one of the popular GNN architectures for materials. We follow Govindarajan et al. (2024) for creating the graphs and crystal skeletons for the MEGNet architecture (we provide additional details in Section B.6). Consequently, the environment can be easily customized to incorporate other graph- and non-graph-based policy networks. For each agent, in each setting, we train five different seeds.

### 5.1    Feasibility of RL-based approaches

To assert that RL-based methods can indeed generate high-quality crystals in terms of desired properties, we select the simplest variation of the different benchmark settings, where the agent trains on the same crystal structure, optimizes for target values that are in-distribution, and uses the small action-space of 18 elements. This allows us to compare the performance of different RL approaches and identify the properties that are difficult to optimize. We also intend to determine if there exists at least one solution, i.e., composition for each of the seven structures (shown in Figure 8) that correspond to a property value close to the desired target. The performance comparison of PPO, Rainbow, DQN and SAC for each structure and all properties is shown in Figure 3 and Figure 10.

**PPO**    In general, PPO quickly finds an optimal or suboptimal solution after a short period of exploration and converges at that point. This is particularly helpful for mechanical properties like density and bulk modulus that have a less complex reward landscape. However, for band gap, considering the large failure

rate and the tendency of DFT to produce near-zero values, PPO performs poorly – while it learns to avoid failure states, it converges to a value corresponding to a zero band gap, and does not improve thereafter. Although PPO observes high-reward solutions during training, the inherent complexity of the property does not direct the agent toward those useful states.

**DQN & Rainbow**  The exploration in purely value-based methods like DQN and Rainbow follows a $\epsilon-$greedy scheme, unlike PPO. Therefore, the agent starts with a uniform random exploration and gradually exploits the strength of the policy as it learns from more samples from the environment. The samples are temporarily stored in a replay memory during training, which helps the agent process past information in batches and stabilizes learning. For all properties, the learning curve indicates a steady improvement and convergence close to the optimal solution. As expected, band gap, which is the hardest property to optimize, requires additional exploration, resulting in slower learning, and high returns are reached only in some structures. DQN and Rainbow demonstrate similar learning behavior and performance in most cases. However, in some cases, the difference in sample efficiency is visible (e.g. `C2` and band gap in Figure 3).

**SAC**  With SAC, none of the experiments for any structure or property showed a positive learning curve. The agent is unable to escape the exploration phase. Further investigation is needed to determine the cause of the learning issue with SAC, despite using the same hyperparameters as DQN and Rainbow for the value-based components, i.e., buffer size and target network update frequency. Two recent works (Zeni et al. (2025); Asad et al. (2025)) indicated the underperformance of the original implementation of SAC with discrete action space tasks, and suggested ways to mitigate slow learning, which we intend to examine for CrystalGym tasks as a future direction.

## 5.2 Increasing the difficulty of individual settings

We now analyse the impact of increasing the difficulty of each of the 3 axes of the design problem: using out-of-distribution (o.o.d.) targets, a larger action-space, or optimizing on mixed crystal structures. We summarize all results in Figure 6, while all learning curves are available in Section C. Notice the high standard deviation for bulk modulus and band gap results, due to multiple DFT failures or noisy results, as well as significant differences in crystal characteristics.

**o.o.d target values**  With harder target values, the algorithms demonstrated similar learning behavior. The plots for structures `C1-7` are shown in Figure 11. While it is equally easy to reach the harder target values in the case of bulk modulus and density in most cases, achieving a band gap of close to 2 eV is at least as hard as 1.12 eV. In `C2`, only Rainbow has managed to show a favorable learning curve, but it does not reach close to optimality. As mentioned in Section 3.2, DFT is known to systematically underestimate the band gap energy, which makes it more likely to output lower band gap values (Lejaeghere et al., 2016). As seen in the plots, it is extremely rare that the agent explores the higher band gap regions. Hence, amidst the high failure rate of DFT calculations, i.e., negative reward, and frequent occurrence of near-zero band gap states, the agent fails to learn in a sample efficient way from the very few high-reward states it encounters. Therefore, choosing target values in the rarer regions in the property distribution adds additional complexity to the learning algorithm.

**Larger action space**  We aim to see if increasing the action space by including more frequently seen elements and transition metals like Iron (Fe) and Cobalt (Co) drive the agent towards different and diverse solutions, where the focus is again on harder target values. However, this also increases the complexity of the problem and makes exploration harder particularly due to the higher chance of DFT failures – the presence of transition metals and heavier elements is likely to cause convergence or charge-related issues in DFT calculations. As seen in the results (Figure 6, middle plot and Figure 12), high returns are easily reached in the case of bulk modulus and density with PPO, DQN, and Rainbow. Band gap computation experiences a significantly higher number of failures, thereby making it harder for the reward to even cross 0.0 for all algorithms. Density optimization again appears to be the easiest of the three tasks. In Figure 7, we show examples of policy-generated crystals (structure `C1`) for hard targets when trained with both small and medium action spaces.

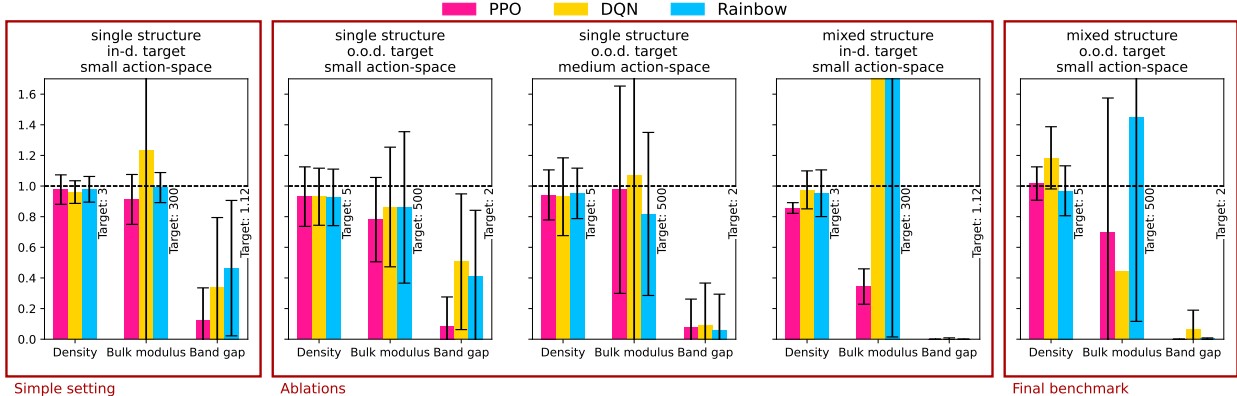

Figure 6: Final performance of each algorithm on each property, for each experiment. After training, each agent is evaluated on 5 trajectories. We report the average achieved property as well as the best-performing algorithms for each setting.

**Mixed crystal structures**  In the next set of experiments, we increase the difficulty of the task, where the goal is to optimize the properties of 5 crystal structures together. As shown in the results in Figure 4, we notice that the algorithms do not reach close to the optimal solution as quickly as in the previous experiment, where the same crystal structure was sampled in every episode. *PPO's exploration and learning strategy seems to remain the same*, but the returns indicate that it reaches a suboptimal policy for all properties. Rainbow and DQN converge to a higher return, indicating that these value-based methods encourage more exploration and learn better even in this difficult task. In the case of band gap, Rainbow and DQN appear to gradually reach high returns, indicating the possibility of reaching optimality with further training despite the complexity of the property and the task of optimizing multiple structures. Finally, similar to the results for the easier tasks in Section 5.1, SAC again demonstrates the poorest learning performance with all three properties.

**Results on the final benchmark**  After analysing the different components and settings of CrystalGym, we train all 4 RL algorithms on our proposed CrystalGym benchmark, which uses the mixed structure, o.o.d targets, and small action-space (Figure 5). First, we notice that, just like for the simpler settings, PPO, DQN and Rainbow can generate crystals with desired density values. The density property serves thus more as a sanity check, as it is relatively simple to achieve, and for which DFT computations are fast. However, the algorithms perform poorly on bulk modulus and band gap. During evaluation, only a few seeds, on a few crystals resulted in non-zero band gap computations, with many of the DFT calculations resulting in failure. This shows the complexity of optimizing crystals for accurate properties, and makes a stark contrast with optimization through ML property predictors.

**Overall analysis**  We show that varying the RL algorithm, property of interest, and task complexity provides an insightful set of analyses and multiple avenues for future directions. In this section, we explain our analysis at the level of each of the three properties we are interested in optimizing. Firstly, the nature of the calculation differs depending on the property. While bulk modulus and density are mechanical properties and are directly influenced by the atomic weights, the former requires multiple single-point calculations for different volume perturbations. These single-point calculations primarily focus on total energy estimation and can also provide an estimate of the mass density. Moreover, the distribution of these properties suggests a good range of values, and even a randomly sampled composition could result in a value within this distribution. Band gap, which is an electronic property, additionally follows a different set of methods in the single-point calculations that resolve the electronic structure of the crystal and estimate the energies of the highest occupied and lowest unoccupied electronic states. For these types of calculations, DFT is known to have significant underestimation issues and can result in inaccurate estimates. As seen in the results, DFT is highly likely to output near-zero values. The unconventional nature of this property makes it harder for

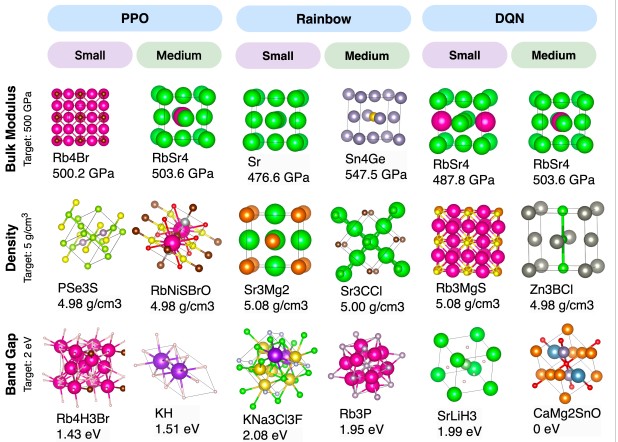

Figure 7: Visualization of top crystals generated for all algorithms on **o.o.d** targets with small and medium action space. Good candidates appear for density and bulk modulus, but band gap remains difficult. Some of the compositions (e.g. $SrLiH_3$) are found in the Materials Project.

RL algorithms to effectively reach a better solution. While higher-order methods exist for more accurate estimation of the band gap values, they are far more time-consuming than regular DFT computations. The scope of this study is limited to unrelaxed crystal structures, considering the significant computational cost involved in structure relaxation with DFT. We further justify our assumption in Section A. As a result of not optimizing the thermodynamic stability, many of the generated structures had positive formation energies, shown in Section B.8.

**Novelty**    We search for solutions directly in the action space rather than learning from a dataset, starting from a completely random policy. Also, given that the structure is fixed in our case, we do not expect any of the generated crystals to exactly match with an existing crystal in public databases. While checking if the reduced composition of the generated crystals exists in Materials Project, we found that over 60% of the compositions were entirely novel (out of 3000 compositions obtained from all experiments). A more comprehensive novelty analysis using the LeMat-GenBench evaluation protocol (Betala et al., 2025) is provided in Section D.3, where we report an overall novelty rate of 89.2% with respect to the LeMat-Bulk reference dataset.

**Uniqueness, Diversity and Stability**    Since RL primarily aims to converge to a policy with a high reward without explicitly encouraging diverse solutions, diversity is not a primary objective of the current work. We generated up to 5 rollouts for each trained policy (for each seed) with a random initial state, i.e., random traversal order for single crystal experiments, and randomizing choice of crystal structure and the order of traversal for mixed crystal optimization. For single crystal experiments, randomizing the traversal order could result in different solutions even for deterministic policies such as DQN and Rainbow. We assessed the fraction of unique crystal compositions generated across all rollouts (counting all seeds) for each experiment, and found that the average percentage of unique crystals per experiment is 48.5%. A broader evaluation of validity, diversity, stability, and novelty using the standardized LeMat-GenBench protocol (Betala et al., 2025) is provided in Appendix D.3. After evaluating the stability of the generated crystals with a Orb-v3 (Rhodes et al., 2025), a state-of-the-art MLIP, we find that none of them were stable ($\leq 0$ eV/atom energy above the convex hull) or metastable ($\leq 0.1$ eV/atom energy above the convex hull). We attribute this to the absence of energy optimization during RL training, and propose incorporating multi-objective rewards – jointly targeting both the desired property and thermodynamic stability, as a promising future direction for this benchmark.

# 6    LLM agent

**Supervised fine-tuning**    In this section, we extend our analysis and leverage large pre-trained autoregressive models for property-driven crystal generation, aligned with DFT-based reward signals. Using **LLaMA-3 8B** (Grattafiori et al., 2024) as the base model, we perform supervised fine-tuning on crystals from a subset of the Materials Project database (MP-20), following a procedure similar to Crystal-Text-LLM (Gruver et al., 2024), with key differences. Since our goal is to generate the composition of crystals given atomic positions, we adopt a different string representation: atom positions are listed first, followed by corresponding elements in order (Figure 21). This is different from the Crystal-Text-LLM representation, where the positions and types of atoms are interleaved. Second, we utilize special tokens for the chemical elements, which allows us to constrain the output to a specific action space. This is particularly useful for our experiments, where we want to limit the action space to a subset of elements from the periodic table. The embeddings for the new tokens are initialized to the embeddings of the corresponding chemical symbol (e.g. H). If the symbol maps to two tokens (e.g. Mn), we initialize with the mean of embeddings of the two tokens. We fine-tune the base model with Low Rank Adaptation (LoRA) (Hu et al., 2022) and 8-bit quantization on the MP-20 training set, which contains $\sim 20K$ crystals. The model is trained for 100 epochs with an effective batch size of 64, AdamW optimizer and cosine learning rate scheduler on 4 H100 GPUs. Similar to Crystal-Text-LLM, we introduce group translations during training, which translates the unit cell by a randomly sampled vector. This introduces an inductive bias that helps the model learn the translational invariance inherent to crystal structures.

**RL fine-tuning**    We use the SFT model as the initial policy for RL fine-tuning. The RL prompt includes lattice parameters and atom positions listed sequentially, and the model is trained to predict the corresponding atom types. For simplicity and practicality, we evaluate REINFORCE only on density and band gap (in-distribution targets) with the smallest action space. All experiments use a batch size of 1, with gradient updates after each episode, and only the LoRA parameters are fine-tuned. During inference, we apply two constraints: 1) output length is fixed to the number of atoms (known beforehand), and 2) generation is limited to element tokens from the small action space. Thus, we stochastically sample $N$ tokens for $N$ atomic sites using top-$k$ sampling, where $k$ is the size of the action space. Coherent with standard practices of fine-tuning LLM with policy gradient approaches, we optimize the policy gradient loss with two regularizers: KL divergence from the initial policy and output entropy. The advantage is set to the reward at the end of each episode.

Our goal is to determine if leveraging large pre-trained language models as policy networks improves the performance and sample efficiency in terms of effectively optimizing for the desired target value. To this end, we train REINFORCE on a few CrystalGym tasks. Results are shown in Section E.

**Single crystal optimization**    We investigate the learning behavior of the LLM policy with REINFORCE when only a single crystal is optimized for a given property. For density, we notice that after a few 100 episodes, the policy converges to a near-optimal or suboptimal reward, and subsequent training does not modify the policy's behavior. With band gap, the frequent DFT failures and frequent 0 band gaps do not help the policy in getting exposed to high rewards. Moreover, as the direction of the policy gradient is determined by the sign of the advantage value, frequent failures (i.e., $-1.0$ reward) push the REINFORCE policy gradient loss down, thereby fooling the policy toward a monotonically decreasing loss, when it does not learn anything meaningful. Also, the entropy loss rapidly drops to zero, leading to convergence toward a deterministic policy that performs worse.

**Mixed crystal optimization**    Given the promising results with density optimization in the single crystal optimization case, we use the SFT policy to optimize the composition of 5 crystals together with the smallest action space for density ($\hat{p} = 3g/cm^3$). Unlike Rainbow and DQN, which reached a high reward with only graph policy networks, the LLM+REINFORCE training pipeline failed to show a positive learning behavior even after 10,000 episodes ($\approx 50,000$ steps). While the KL loss eventually reduces, the entropy and the policy loss do not lean towards convergence to optimality. The failure of a pre-trained LLM policy to learn on the simplest property suggests further investigation is required to determine the best procedure for aligning LLM policies with reward signals from DFT.

## 7  Conclusion

This research aims to take a step in the direction of accelerating the material discovery process, for which performing atomic simulations is inevitable. We show that crystal design is an interesting and useful set of problems dealing with reinforcement learning with expensive reward signals obtained from expensive atomic simulations. Our new environment is modular and allows the addition of different levels of complexity in the tasks, including the choice of the DFT calculator. From an RL perspective, this environment and benchmark boosts research in the direction of learning with expensive and noisy rewards (Wang et al., 2020), and could influence other domains in scientific discovery and beyond. An important limitation of this analysis is not taking into account structure relaxation and the diversity of generated materials. Classical reinforcement learning aims to maximize the expected return and converge to a single behavior. Further investigation of entropy-based RL methods and GFlowNets (Bengio et al., 2021) on these environments is a promising future direction. Other major avenues to explore are 1) learning on large crystal structures with diverse symmetric properties, 2) multiobjective RL for optimizing multiple properties, and 3) introducing new useful properties like shear modulus.

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

# A    Assumptions

**Feasibility of a solution**    In all the experiments for single structure optimzation, we assume that, given a target property $\hat{p}$, there is at least one composition for the structure that can be achieved with the given action space, and results in a property value $p$, where $|p - \hat{p}| < \delta$. Here $\delta$ is assumed to be practically small, and can vary depending on the structure chosen to be optimized.

**Fidelity of reward function**    The current version of the CrystalGym environment uses Quantum Espresso, a software suite for performing DFT calculations, and the signals obtained from DFT are used to compute the rewards for the RL agent. There are several approaches to improve the accuracy of DFT calculations or use higher-order methods for estimating challenging properties like band gap. However, such approaches are expected to take orders of magnitude more time than the current configuration of DFT we rely on for our experiments, that works for most practical cases. Hence, during policy learning, we assume that the signals obtained from the chosen DFT configuration is functionally the highest fidelity we can observe. Consequently, the reinforcement learning workflow aims to directly optimize for the scalar values obtained from DFT calculations.

**Crystal validity**    In standard chemical discovery tasks, it is a common practice to report the percentage of valid candidates generated by the model. The criteria for validity for small molecule discovery is usually molecules following appropriate valency rules. For crystals, the structural and compositional validities are generally measured, where the former deals with the closeness of two atomic sites in a crystal unit cell, and the latter checks if the total charge adds to zero. However, we directly rely on the outputs of DFT, which is expected to fail to simulate or converge for theoretically infeasible crystals.

**Structure relaxation**    For practical reasons, we do not perform structure relaxation on policy-generated crystals. Although DFT relaxation optimizes the crystal structure to minimize system energy, which benefits downstream applications, it requires multiple single-point DFT calculations per sample, significantly increasing computational complexity. Hence, the backbone structure and lattice of each crystal candidate in the environment is unchanged during training and evaluation. As an immediate next step, we wish to explore the idea of including machine learning potentials in the RL loop for faster structure relaxation.

# B    Experimental details

Note: Some runs appear truncated due to the high computational cost of DFT-based RL training. Extended results will be included in the revised version.

## B.1    Action space

The CrystalGym environment allows the possibility of using different action spaces. The scope of this benchmark is limited to action spaces corresponding to two sets of elements from the periodic table. The smallest action space contains 18 elements, which are mostly Group 1 and Group 2 metals and some nonmetals, but no transition elements. This **small** action space simplifies DFT calculations, resulting in lesser number of failures in simulations, but vastly reduces the exploration space compared to the ideal action space (118 elements in the periodic table). The **medium**-sized action space consists of some of the frequently occurring transition metals, in addition to the elements in the smaller action space. We also propose a **larger** action space that includes rarer transition elements, which we aim to test in the future.

- **Small**: Li, Na, K, Rb, Be, Ca, Mg, Sr, H, C, N, O, P, S, Se, F, Cl, Br

- **Medium**: Li, Na, K, Rb, Be, Ca, Mg, Sr, H, C, N, O, P, S, Se, F, Cl, Br, B, Si, Ge, Fe, Cu, Co, Ni, Mn, Al, Zn, Sn, Cr

- **Large**: Li, Na, K, Rb, Be, Ca, Mg, Sr, H, C, N, O, P, S, Se, F, Cl, Br, B, Si, Ge, Fe, Cu, Co, Ni, Mn, Al, Zn, Sn, Cr, In, Sb, V, Mo, Ga, Ag, Ti, Ba, Y, Te, I, Pd, Rh, As, Pt, Cs, Au, Bi, Zr, La

Table 1: List of the different properties values for the **easy** and **hard** settings.

| | **Bulk Modulus** (GPa) | **Density** ($g/cm^3$) | **Band Gap** ($eV$) |
|---|---|---|---|
| **Easy** (*in-d.*) | 300.0 | 3.0 | 1.12 |
| **Hard** (*o.o.d.*) | 500.0 | 5.0 | 2 |

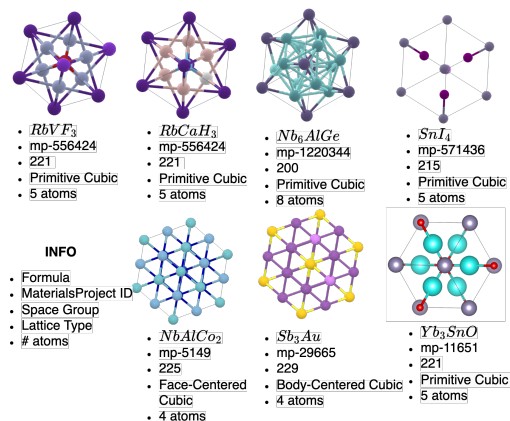

Figure 8: Every possible starting crystal structure considered for our experiments. Each structure has been picked from existing crystals in the Material Project database and their corresponding geometric properties are displayed below their representation. From left to right and top to bottom, the crystals have been referred to as C1, C2, C3, C4, C5, C6 and C7 in the paper.

## B.2 Environment variations

The CrystalGym environment allows testing RL algorithms on a variety of tasks with customizable levels of difficulties. The list of variations supported in the current version of the environment is shown in Table 2.

Table 2: List of all the variations of experimental components. Each experiment is designed to study the impact of specific variations across different configurations of experimental components.

| **Experiment** | **Variations** |
|---|---|
| RL Algorithm | PPO |
| | SAC |
| | DQN |
| | Rainbow |
| Properties | Density |
| | Bulk Modulus |
| | Band Gap |
| Structures | Single |
| | Mixed |
| Mode | Completion |
| | Substitution |
| Policy Net | MEGNet |
| | CHGNet |
| Action Space | 18 |
| | 30 |
| | 50 |

### B.3 Target properties

For our experiments, we use two sets of target values, one being *in-distribution* and other being *out-of-distribution*. The values are listed in Table 1. For each of the three properties, distributions of the values for **all cubic crystals** in the Materials Project database are shown in Figure 9.

**Bulk modulus**   The bulk modulus distribution shows that the mode falls between 100-150 GPa. Our chosen *easy* (i.e., *in-d.*) target of 300 GPa is in the rarer regions in the distribution, but there is a reasonable number of crystals that have a bulk modulus of close to this target. However, the target of 500 GPa exists outside this distribution, indicating that it could be a hard value to reach through exploration.

**Density**   The distribution of densities shows that both the chosen target values lie well within the distribution. However, density is directly related to the total mass of the crystal, which is dependent on the atomic weights. Hence, it is mostly the choice of the action space that determines how easily the agent can reach higher density values. While in most cases the agents could reach 5 $g/cm^3$ easily, our separate analysis of PPO's performance with a target of 8 $g/cm^3$ highlighted failure of the agent to reach densities close to optimality for all crystal structures (Figure 14).

**Band gap**   The plot of band gap frequency shows a highly skewed distribution, where majority of the crystals have a band gap value close to zero. Both the *easy* (1.12 eV) and *hard* (2 eV) targets lie in the rarer regions of the distribution. However, with the type of DFT calculations we perform with Quantum Espresso, it can be noticed that it is extremely rare that the agent experiences states with band gaps higher than 1.5 eV during training. This makes 2 eV much harder as a target than 1.12 eV.

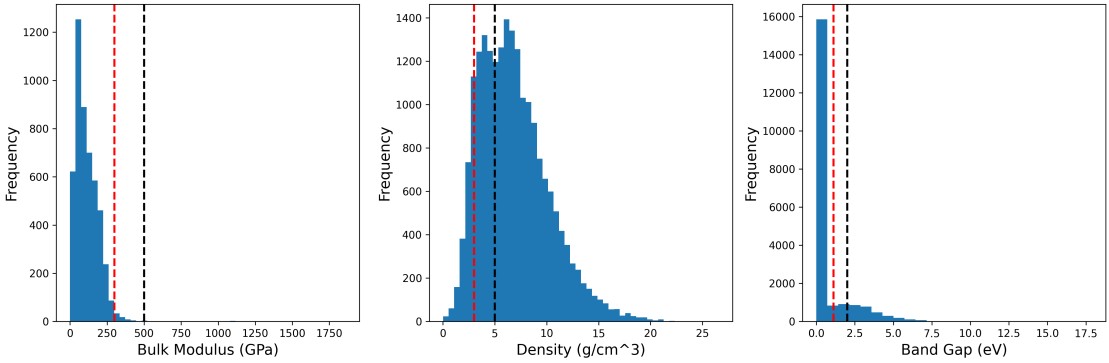

Figure 9: Histograms of the distribution of Bulk Modulus, Density and Band Gap for crystals in the Material Project database. The dashed red line represents the value chosen for the **easy** target and the black for the **hard** one.

### B.4 Reward functions

The reward functions were chosen based on the type and range of the properties of interest. For instance, bulk modulus can take a wide range of values, and since exponential distance would hugely amplify small deviations, we chose to use the absolute distance function – the reward is therefore the negative absolute distance. Since the reward is always negative for bulk modulus, we decided to clip it to a minimum value of -5 to avoid large negative rewards. Table 4 shows further details of the reward functions for each property including the bounds and computation times.

Table 3: Experimental setup detail. Experiments 1-5 generate crystals from scratch while experiment 6 replaces atoms in fully completed crystals. Each experiment has a unique combination of action space size, target value of the property and size of the pool of starting crystal structures.

| Exp No. | Mode | Target | Action Space |
|---------|------|--------|--------------|
| **Completion** | | | |
| 1 | Single | Easy | Small |
| 2 | Single | Hard | Small |
| 3 | Single | Hard | Medium |
| 4 | Mixed | Easy | Small |
| 5 | Mixed | Hard | Small |
| **Substitution (CHGNet)** | | | |
| 6 | Mixed | Easy | Small |

Table 4: Properties of the reward functions used for each property. In addition to the mathematical details of the normalization used we provide some important DFT-specific characteristics.

| | **Bulk Modulus** | **Density** | **Band Gap** |
|---|---|---|---|
| Reward Formulation | Absolute distance | Exponential distance | Exponential distance |
| Max. Reward | 0.0 | 1.0 | 1.0 |
| Min. Reward (failure) | -5.0 | -1.0 | -1.0 |
| Range | [-5,0] | $\{-1\} \cup (0,1]$ | $\{-1\} \cup (0,1]$ |
| Time (s) | $\approx 130$ | $\approx 20$ | $\approx 20$ |
| Failure Rate (%) | $\leq 0.1$ | $\leq 0.01$ | $> 20$ |

### B.5 DFT settings (Quantum Espresso)

We performed DFT single-point SCF simulations using Quantum Espresso v7.1 (Giannozzi et al., 2009), which is fully open-source. Solid-state pseudopotentials from SSSP version 1.3.0 (Prandini et al., 2018) were used for the calculations. The settings used are listed below.

1. `calculation`
   - `scf` for band gap and bulk modulus
   - `vc-relax` for density

2. `nstep`: 1

3. `nbnd`: $\left\lceil \left( \left( \sum_i^N Z_i \right) \mathrm{div}\, 2 \right) * 1.2 \right\rceil$

4. `ecutwfc`: 50

5. `ecutrho`: 400

6. `occupations`
   - `smearing` for bulk modulus and density
   - `fixed` for band gap

7. `degauss`: 0.001

8. `nspin`: 1

9. `electron_maxstep`: 300

10. `mixing_mode`: plain

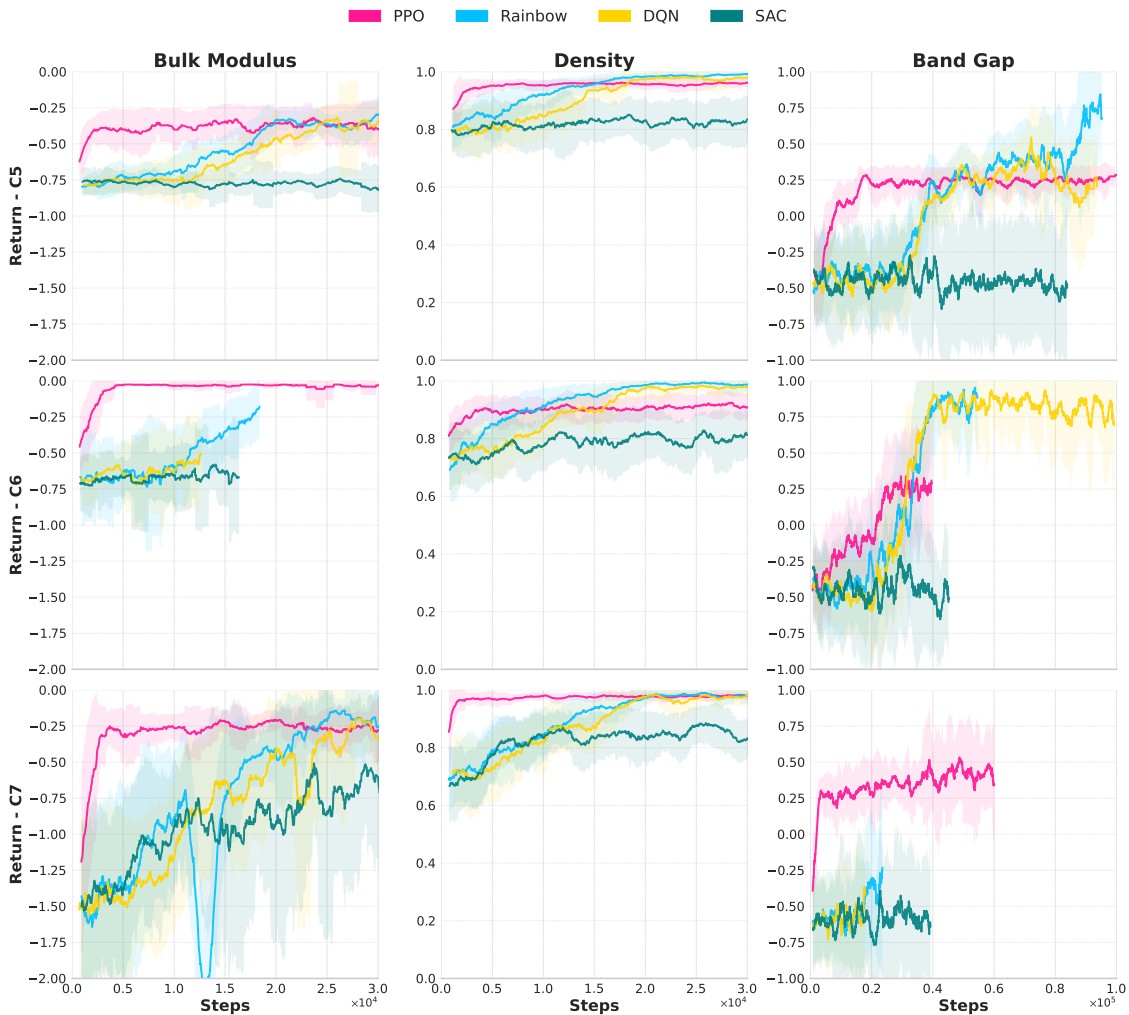

Figure 10: Training curves for crystals C5, C6 and C7 for Experiment 1, with a **single** starting structure, a **small** action space and ***in-d.*** targets.

11. `mixing_beta`: 0.7

12. `diagonalization`: david

13. `kpoints`: Chosen automatically from Kpoint density using Pymatgen (Ong et al., 2013).

## B.6 GNN details

In order to extract meaningful representations from crystal structures, we chose to use graph neural networks conditionned on the target property in every algorithm. These representations are then fed to projection layers to compute each algorithm's relevant quantities. DQN and Rainbow use this arcihtecture as their Q-networks and PPO and SAC use it for both their value and policy networks. In each case, we only need to adapt the MLP's output shape.

### B.6.1 MEGNet

MEGNet (Chen et al., 2019) is a universal graph machine learning framework for molecules and crystals that provides expressive graph representations through a message passing scheme specifically tailored for crystals and molecules. We used MEGNet as the default GNN architecture in our experiments. It takes as

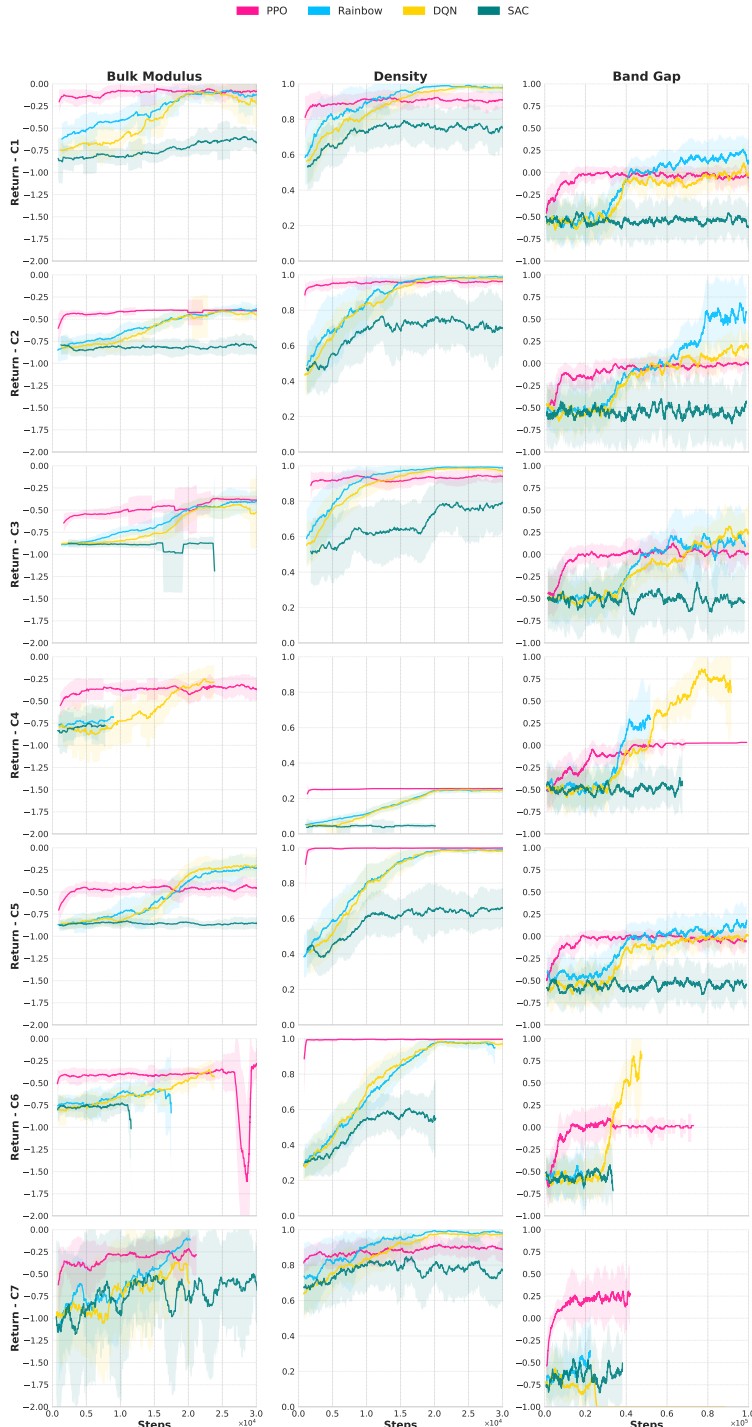

Figure 11: Training curves for all crystals for Experiment 2, with a **single** starting structure, a **small** action space and **o.o.d.** targets.

input a graph $\tilde{\mathcal{G}}(\tilde{\mathbf{H}}, \tilde{\mathcal{I}}, \tilde{\mathbf{y}}; \hat{p})$, where $\tilde{\mathbf{H}}$, $\tilde{\mathcal{I}}$, $\tilde{\mathbf{y}}$ and $\hat{p}$ are respectively the embeded node features, the embeded edge features, the embeded graph-level features and the target property the model is conditioned on. The categorical node features $\mathbf{H}$ are defined as the one-hot encoding of the atom type for each node of the graph, with an additional dimension indicating whether the node is filled with an atom or not. They are then

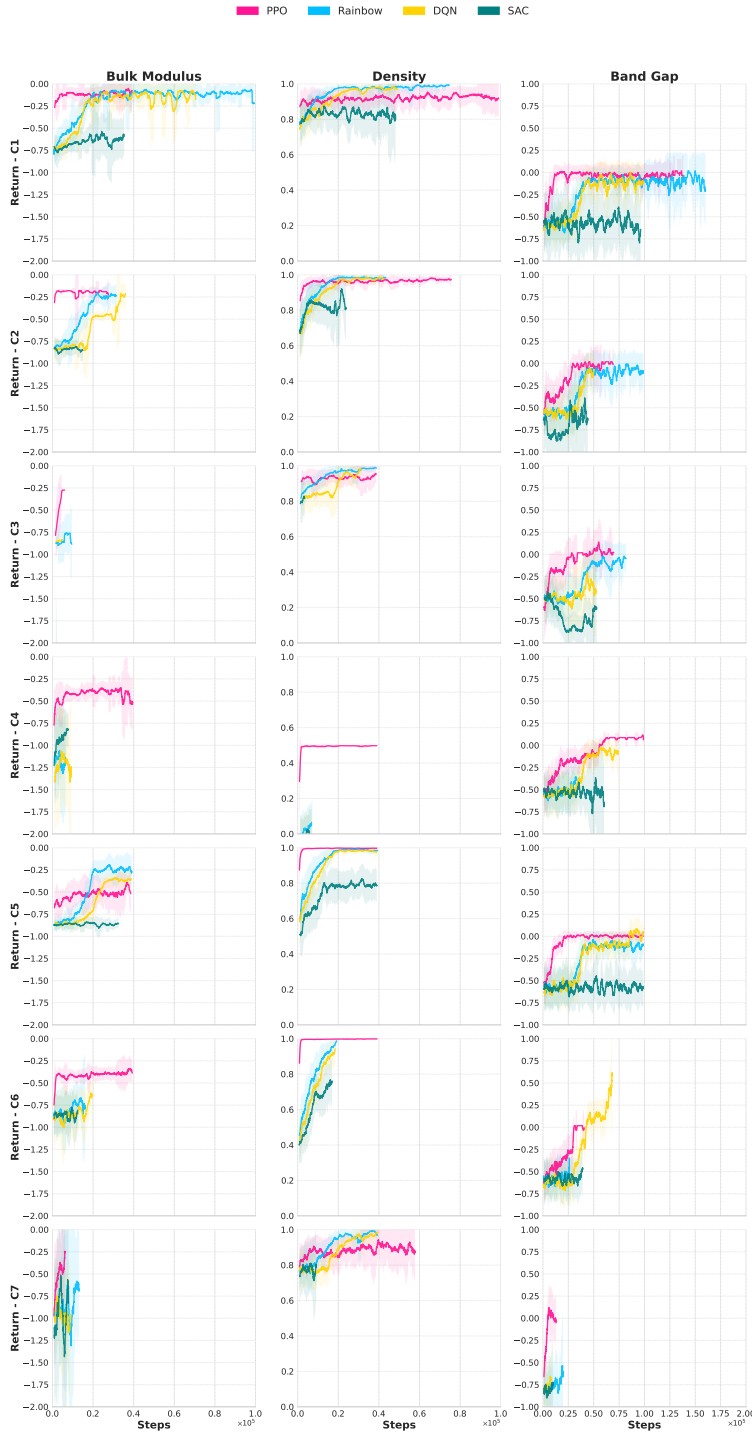

Figure 12: Training curves for all crystals for Experiment 3, with a **single** starting structure, a **medium** action space and **_o.o.d._** targets.

passed through embeding layers to obtain $\tilde{\mathbf{H}}$. Edges connect neighbouring atoms based on the CrystalNN scheme (Pan et al., 2021) for determining their type and presence. We derive the edge features $\mathcal{I}$ as the set $\{t_{uv,(c_1,c_2,c_3)}\}$ of gaussian distances between atoms $u$ in the reference unit cell and $v$ in a unit cell shifted by $c_1\mathbf{l_1} + c_2\mathbf{l_2} + c_3\mathbf{l_3}$.

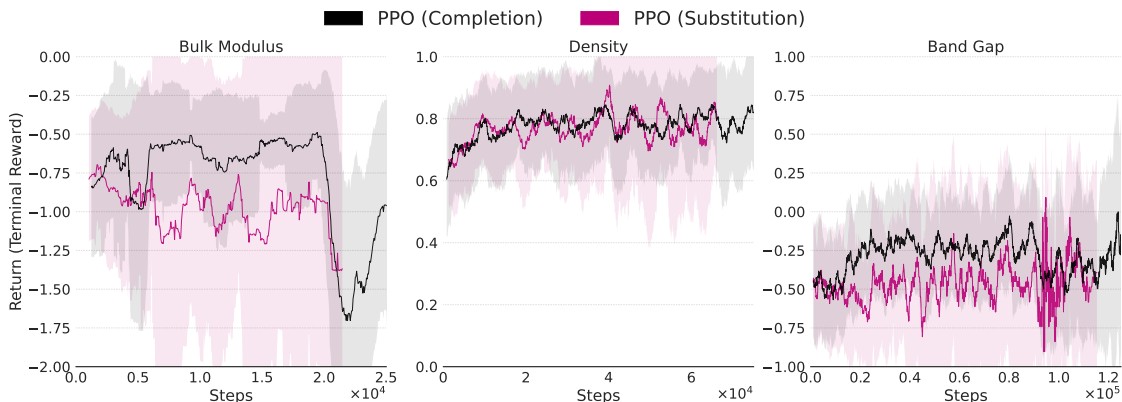

Figure 13: Training curves for Experiment 6. This experiment compares the differences between the completion and substitution approaches on crystal C1 with PPO. The experimental configuration has **mixed** starting structures, a **small** action space and **in-d.** targets.

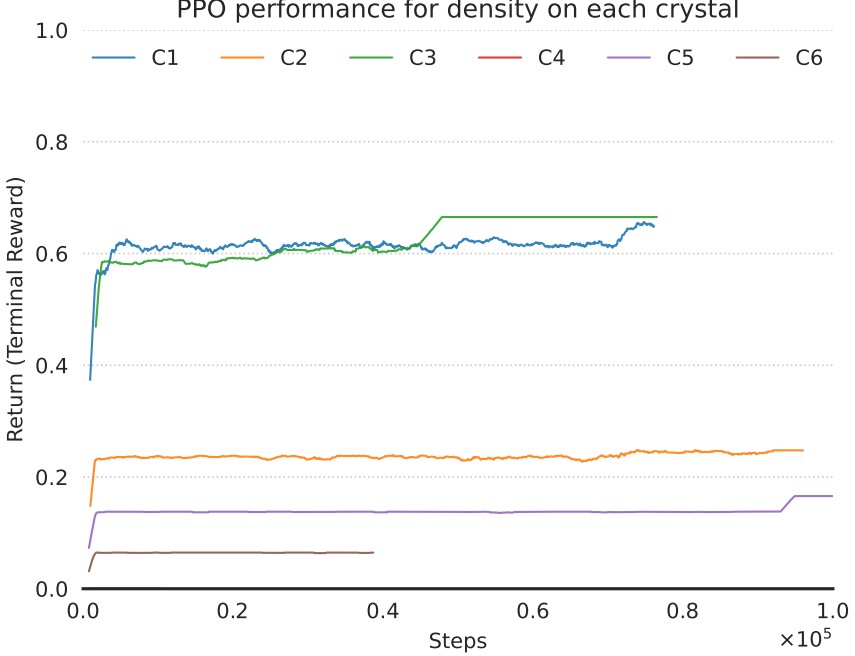

Figure 14: Training curves of PPO for crystals C1, C2, C3, C4, C5, C6, C7. This experiment studies the ability of the agent to generate crystal with a **very hard** target density of 8 $g/cm^3$, a **single** starting structure and a **small** action space.

$$t_{uv,(c_1,c_2,c_3)} = \exp\left[-\frac{d_{uv,(c_1,c_2,c_3)}^2}{\rho}\right] \tag{1}$$

$$d_{uv,(c_1,c_2,c_3)} = \sqrt{\left(\mathbf{x_v} + c_1\mathbf{l_1} + c_2\mathbf{l_2} + c_3\mathbf{l_3} - \mathbf{x_u}\right)^2} \tag{2}$$

where $\mathbf{x_v}, \mathbf{x_v} \in \mathbb{R}^3$ are the 3D coordinates of atoms $u$ and $v$ respectively in the reference unit cell. These edge features are then passed through MLP layers to obtain $\tilde{\mathcal{I}}$. Finally, the graph-level features are defined as $\mathbf{y} = [a, b, c, \phi_1, \phi_2, \phi_3, S, \hat{p}, \tilde{\mathbf{f}}]$ where $a$, $b$ and $c$ are the lengths of the edges of the lattice ($a = \|\mathbf{l_1}\|$, $b = \|\mathbf{l_2}\|$ and $c = \|\mathbf{l_3}\|$), $\phi_1$, $\phi_2$ and $\phi_3$ are the angles of the lattice, S is the space-group number, $\hat{p}$ is the target

property the model is conditionned on and $\tilde{\mathbf{f}}$ is the embeding of the one-hot vector of the categorical feature $\mathbf{f}$, called focus, which instructs the policy which node is to be filled next. $\mathbf{y}$ is then passed through MLP layers to obtain $\tilde{\mathbf{y}}$.

The MEGNet architecture consists of taking as input the graph $\tilde{\mathcal{G}}^{(0)} = \tilde{\mathcal{G}}$ of embeded node, edge and graph-level features and applying $K$ MEGNet layers to it, followed by a readout layer designed to obtain graph-level representations. The Q-values (or values or logits) are obtained by feeding these representations to a MLP layer.

$$\tilde{\mathcal{G}}^{(k+1)} = \text{MEGNet}\left(\tilde{\mathcal{G}}^{(k)}\right) \ \forall \ 0 \leq k \leq K - 1 \tag{3}$$

$$\psi\left(\tilde{\mathcal{G}}^{(K)}\right) = \text{Readout}\left(\tilde{\mathcal{G}}^{(K)}\right) \tag{4}$$

$$Q_\theta\left(\mathbf{s} = \mathcal{G}; \hat{p}\right) = \text{MLP}\left(\psi\left(\tilde{\mathcal{G}}^{(K)}\right)\right) \tag{5}$$

### B.6.2 CHGNet

CHGNet (Deng et al., 2023) is a state of the art graph neural network for modeling a universal potential surface. It is a pretrained model designed to provide rich representations for molecules and crystals. It preserves translation, rotation and permutation invariance of its inputs and has a more complicated process to generate its inputs from the graph of the crystal, namely it considers the graph of atoms and their different bonds as edges, as well as the graph of bonds and their relative angles as edges. Its inputs features include a Fourier representation of the angle information in addition to the regular edge (bonds) and node (atoms) features. The CHGNet layer is applied $K$ times just like for MEGNet, but its message passing function is more complex, allowing for deeper interactions between the node, edge and angle informations. We replaced the energy prediction layer by an uninitialized MLP to output Q-values, state values or logits depending on the algorithm used. We froze the weights of the CHGNet as the network is pretrained and provides good representations and only trained the MLP layer we added.

### B.7 SAC hyperparameter tuning

We performed a hyperparameter search for SAC, and none of the configurations gave a promising performance. The hyperparameters we tuned were: target network update frequency, replay buffer size, and temperature. Figure 15 shows the hyperparameter tuning experiments for one of the crystal structures, where bulk modulus (300 GPa) is the target property. Figure 16 shows the temperature tuning experiments, where the target property is 3 $g/cm^3$ density. While pure value-based approaches (DQN, Rainbow) that use target nets and replay buffers work fine in multiple cases, we believe that the learning scheme of SAC does not allow the agent to escape the exploration phase. It needs to be investigated if this problem is well-suited for SAC.

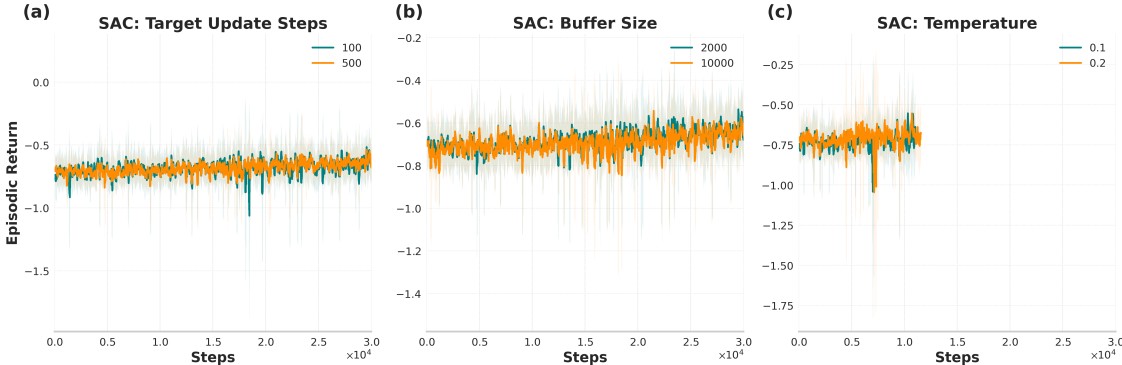

Figure 15: Hyperparameter experiments with SAC – optimization with single structure for **300 GPa bulk modulus**. Tuning of (a) target network update frequency; (b) replay buffer size, and (c) temperature parameter.

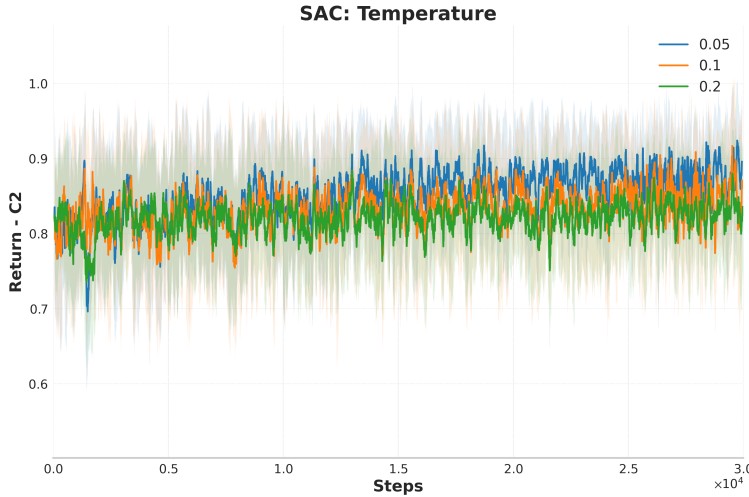

Figure 16: Hyperparameter experiments with SAC – optimization with single structure for **3** $g/cm^3$ **density**. Tuning of temperature parameter (alpha).

### B.8   Formation energy analysis

We computed the formation energies of the policy-generated crystals using DFT. Following (Govindarajan et al., 2024), we first evaluated the total energies of the elemental reference crystals, which were then used to calculate the formation energies. As a result of not relaxing the crystals and not optimizing compositions for stability, the formation energies of many of the crystals were positive, and the distribution is shown in Figure 17. While our simplified design choices helped us test RL algorithms specifically focusing on the three properties of interest individually, we intend to incorporate structure relaxation and energy optimization as future work. As a result, we do not claim that the RL-generated crystals are stable or easily synthesizable.

## C   Substitution

In all the previous experiments, we focused on completing the backbone of a crystal structure, where the initial state does not have any atoms filled, and the intermediate states are partially filled crystals. In this experiment, we intend to determine if using a large pre-trained physics-based graph neural network (GNN) trained on crystalline materials could serve as an effective initial policy. However, with completion, the intermediate states are invalid crystals, and cannot be directly used with these GNNs. We instead focus on

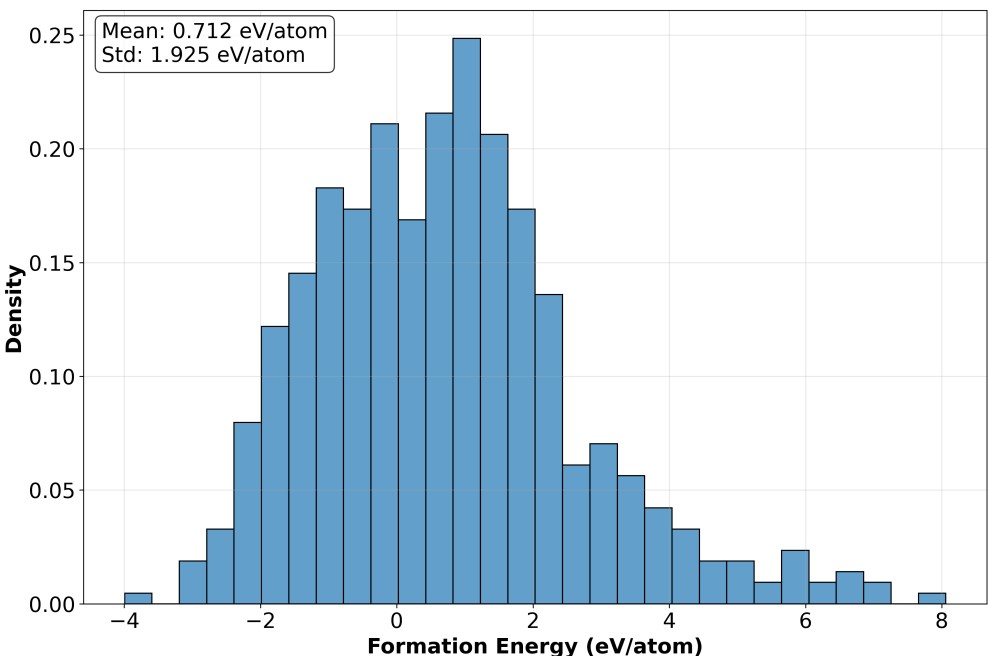

Figure 17: Histogram of the formation energies of around 1000 of the RL-generated crystals, obtained from policies trained for different tasks and properties.

substitution, where the agent substitutes an atom in a given atomic site at each step. In such a case, the initial state would be a potentially valid crystal with randomly placed atoms in all the atomic sites. The intermediate states can also be rendered into a valid crystal, making it easier to pass them as inputs to state-of-the-art pre-trained GNNs. These networks are then subsequently fine-tuned with the RL objective. As the scope of this analysis is limited to the policy network architecture, we only investigate the performance of PPO for optimizing each of the properties with a pre-trained CHGNet backbone model as the initial policy. The results indicate no favorable performance with the larger and pre-trained policy. This further suggests that the complexity of the problem is primarily tied to the nature of the reward signals.

# D    Additional Analysis

## D.1    Area Under the Curve (AUC) analysis

To quantify sample efficiency, we computed the area under the return curve (AUC) for each algorithm, which provides a single scalar summary of both the speed of learning and the quality of the converged policy. We present results for crystal C2 in the simplest experiment, where a single structure is optimized for "easy" *in-d.* target values, given the smallest action space.

These results are consistent with the qualitative observations in the paper (for single structure optimization): PPO achieves higher sample efficiency for bulk modulus and density, where the reward landscape is smoother and DFT failure rates are low, while Rainbow performs best for band gap, where its experience replay helps it learn from the rare high-reward states encountered during exploration. SAC consistently shows the lowest AUC across all properties, corroborating our finding that it struggles to escape the exploration phase in this setting. We will include AUC analysis across all crystals and experiments in the revised version to provide a more systematic and quantitative characterization of algorithmic behavior.

Table 5: AUC values for each algorithm on crystal C2 in simplest CrystalGym benchmark (**single** structure, **_in-d._** targets, **small** action space).

| Algorithm | Bulk Modulus ($\times 10^4$) | Density ($\times 10^4$) | Band Gap ($\times 10^4$) |
|---|---|---|---|
| PPO | **14.7** | **5.9** | 11.3 |
| Rainbow | 13.9 | 5.7 | **11.6** |
| DQN | 13.8 | 5.7 | 10.5 |
| SAC | 12.9 | 5.4 | 5.6 |

### D.2  Conservative Objective Models (COM)

We conducted additional experiments exploring a model-based RL approach, where a learned reward model replaces DFT calls during training and is periodically fine-tuned with DFT-queried data. Concretely, we trained a reward model using the Conservative Objective Model (COM) objective (Trabucco et al., 2021), adapting it to the discrete action space of atom types, building on ideas from (Reddy et al., 2024). The details of this experiment are provided below.

**Hypothesis**  Model-based RL with COM can facilitate faster and efficient learning from DFT signals to produce desirable crystals.

**COM-pretraining**  First, a publicly available graph network, MEGNet for band gap prediction (from https://github.com/materialyzeai/matgl), originally trained on the Materials Project dataset, was loaded. A published dataset with band gap values of Materials Project crystals calculated using Quantum Espresso (from Govindarajan et al.) was then obtained. Next, a reward model was trained with the COM objective, where optimization happens in the discrete space of atom types. For this step, ideas were borrowed from Reddy et al. (2024), as well as the original COM implementation (Trabucco et al., 2021): 1) optimizing for discrete entities by performing gradient ascent starting from the one-hot encoding space and taking the argmax of the resulting solution, and 2) hyperparameters like the number of gradient ascent steps, learning rates for all the optimizers, auto-tuning, etc. Finally, the model with low validation loss (epoch 48) was chosen, and the training curves, including step-wise data for each loss term, are shown in Figure 18.

**Model-based RL**  The trained COM model in the above experiment replaces the DFT-based reward in our original RL setup, and for the sake of demonstration, we only train DQN agents on 4 crystal structures (C1, C2, C3 and C4). Once every 50 steps, we query the DFT oracle and fine-tune the reward model with a single example for multiple gradient steps and a reasonable learning rate. Whenever DFT is called, we log the true reward value and the band gap value.

**Results**  The COM-based reward model quickly collapses as it gets fine-tuned (Figure 19), and the true band gap value is far from converging to the target of 1.12. Figure 20 shows the returns when DQN is trained only with the reward model with no fine-tuning; while the agent learns, after evaluating the trained policy with DFT, the simulation either resulted in 0.0 band gap or failure.

**Conclusion**  The nature of the noise here is different, and it requires carefully developed domain-driven methods. Much of this depends on the machine learning model's capacity to estimate complex properties accurately, and designing such powerful physics-based networks for accurate property prediction is still an active area of research.

### D.3  LeMat-GenBench Metrics

We performed novelty, stability and diversity analysis for 1080 generated crystals with unique compositions using the recently proposed LeMat-GenBench benchmark (Betala et al., 2025), a unified framework for evaluating generative models of crystalline materials. This framework proposes standardized protocols for evaluating the validity, stability, novelty, uniqueness, and diversity of generated crystals. Around 63.6 %

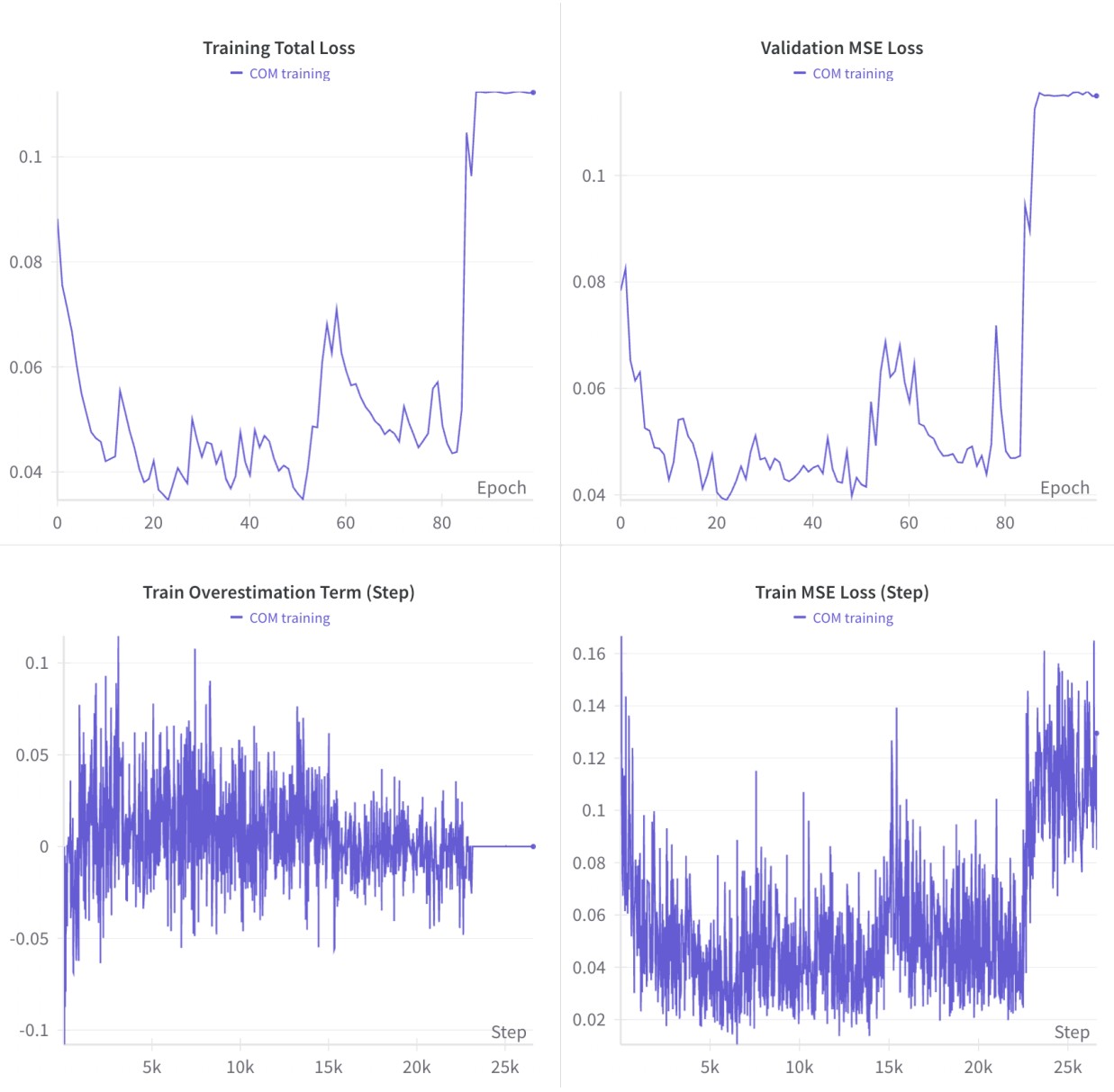

Figure 18: **Loss curves of training a reward model with COM objective**: **a)** Total training loss after every epoch; **b)** Total validation loss after every epoch, **c)** Average overestimation after every train step, and **d)** MSE loss after every train step.

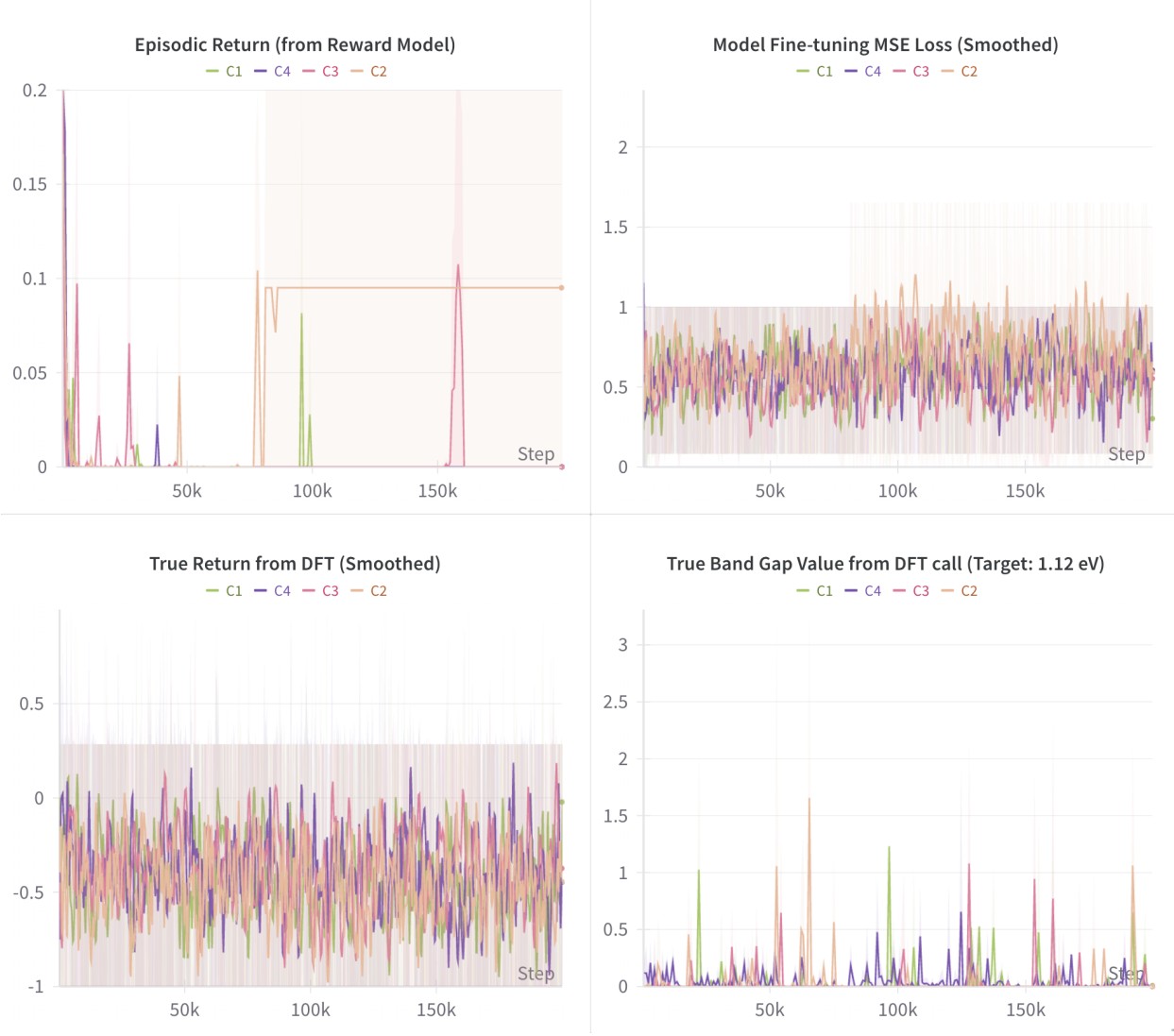

Figure 19: **Training DQN agent separately on 4 crystals (C1,C2,C3,C4) with the COM-based reward model trained in above experiment. The reward model is fine-tuned once in 50 episodes. The target band gap here is 1.12 eV, which corresponds to a maximum reward of 1.0. a) Episodic return from the reward model, b) Loss of fine-tuning reward model after every DFT call, c) Returns from DFT outputs, and d) Logged band gap after each successful DFT call.**

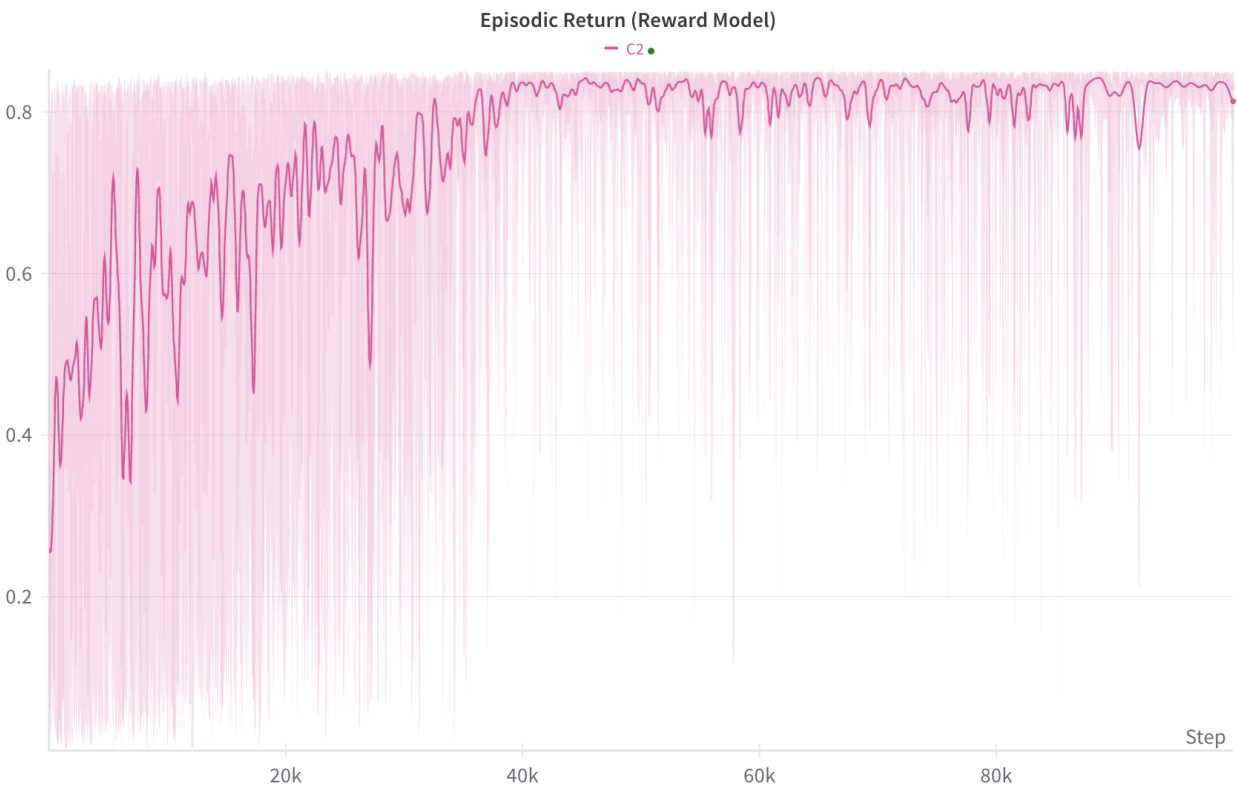

Figure 20: **Training DQN on C2 with the COM-based reward model, but *without* fine-tuning with DFT signals.**

of the structures were valid, where validity encompasses charge neutrality, interatomic distances, physical and plausibility. Novelty is computed with respect to the LeMat-Bulk reference dataset (Siron et al., 2025), which unifies Materials Project, Alexandria, and OQMD (Ghahremanpour et al., 2018; Saal et al., 2013), . Diversity is characterized by the element and space group coverages. Stability and metastability are computed by determining the energy above the convex hull from the Orb-v3 model (Rhodes et al., 2025), a state-of-the-art machine learning interatomic potential. In Table 6 We note the high novelty and attribute this to the fact that our pipeline does not fully rely on training on a static dataset of crystals (in our case, only the skeleton structure is taken from crystals in the Materials Project). However, given that we include only certain elements from the periodic table in the action space, and that our seed structures offer limited variability in the space group, the diversity scores are low. Further, since we do not explicitly optimize for the energy, the stability and metastability rates are 0.0. Achieving higher stability rates is a challenging problem and most of the generative models evaluated with LeMat-GenBench offer very low ($< 5\%$) stability rates.

| Benchmark | Key Metric | Score |
|-----------|-----------|-------|
| **Validity** | Overall valid ratio | 63.6% (687/1080) |
| **Novelty** | Not in LeMat-Bulk | 89.2% |
| **Diversity** | Element coverage | 25.4% |
| **Diversity** | Space group coverage | 10.0% |
| **Stability** | Stable (ORB, e_above_hull=0) | 0% |
| **Stability** | Metastable (ORB, <0.1 eV/atom) | 0% |
| **Stability** | Mean e_above_hull (ORB) | 1.835 eV/atom |

Table 6: Overall Summary (all benchmarks)

# E LLM case study

```
Below is a description of a bulk material with 12 atoms. Given the lengths and angles of the unit cell in
angstroms, and the fractional coordinates for each of the 12 atoms, generate the element type for each atom within
the lattice. The sequence of elements follows the same order as the listed coordinates.:
8.0 8.0 5.6
72 72 21
0.051 0.285 0.348
0.717 0.952 0.346
0.386 0.620 0.342
0.218 0.452 0.847
0.544 0.778 0.840
0.885 0.119 0.852
0.645 0.880 0.036
0.299 0.533 0.045
0.976 0.211 0.057
0.788 0.023 0.652
0.469 0.703 0.638
0.133 0.367 0.653
<elem_Na><elem_Na><elem_Na><elem_Mn><elem_Co><elem_Ni><elem_O><elem_O><elem_O><elem_O><elem_O><elem_O>
```

Figure 21: Prompt used for LLM experiments.

The prompt used for fine-tuning the LLM is shown in Figure 21. The learning plots for the experiments done in Section 6 with density and band gap target properties are shown in Section E and Figure 23, respectively. The results demonstrate poor learning behavior, especially with mixed crystals. Band gap optimization fails even with single crystals.

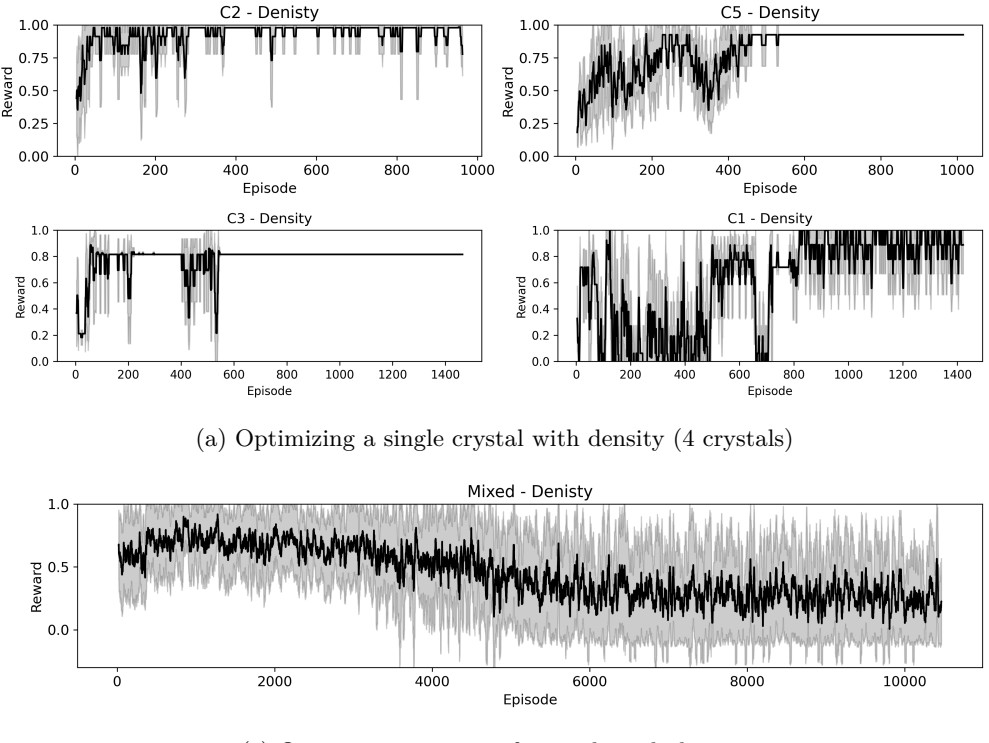

(a) Optimizing a single crystal with density (4 crystals)

(c) Optimizing mixture of crystals with density

Figure 22: REINFORCE with SFT-trained LLM Policy for crystal generation given small action space and density target.

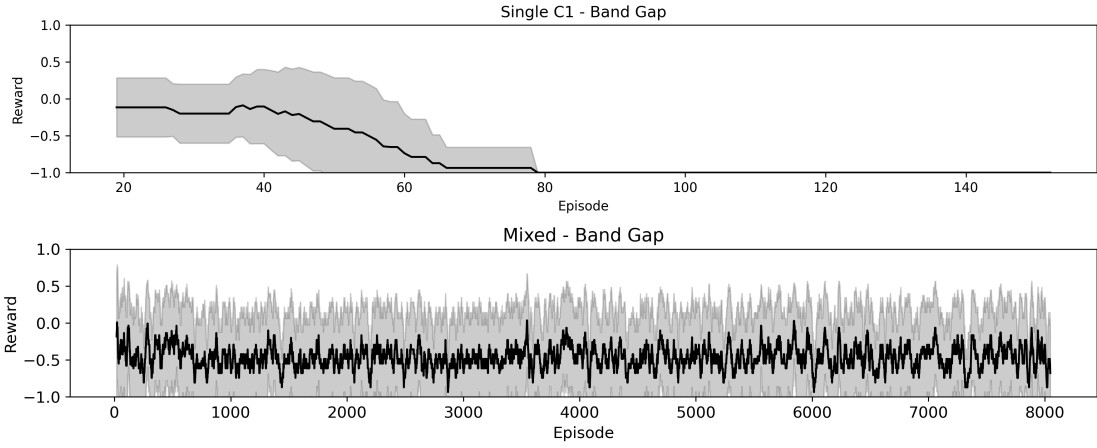

Figure 23: Top: REINFORCE with SFT-trained LLM Policy for crystal generation given small action space and band gap target. Top: Single Crystal. Bottom: Mixture

