# OpenReview forum: "CrystalGym: A New Benchmark for Materials Discovery Using Reinforcement Learning"
_TMLR — Rejected by TMLR_

### Review · Reviewer_zyuu · 2025-12-03

**Summary Of Contributions:**

The authors propose a remarkable open-source benchmark that allows researchers to utilize reinforcement learning (RL) approaches for in silico design and optimization. It also addresses a critical need for performant solutions whereas those that rely on high-accuracy atomic silmulators and density functional theory (DFT) struggle with high computational cost. DFT calculations require significant expertise in quantum mechanical laws.

The authors' framework allows the direct adoption of DFT signals with RL in crystalline material discovery. These problems in this benchmark are extremely difficult to solve, as none of the explored methodologies were able to solve all the tasks, it demonstrates a critical opportunity and need to publish this for RL practictioners as clearly there remains a gap between existing state-of-the-art RL (e.g., PPO, SAC, Rainbow) and real-world problems.

Although I am not an expert or familiar with crystalline materials, the authors do a phenomenal job in describing the essentials of crystalline materials for which I am very grateful.This is a key and necessary aspect in conveying these task scenarios to the TMLR audience.

Worth mentioning that the authors use the latest state-of-the-art optimization strategy, too (e.g., AdamW), demonstrating they are up-to-date in machine learning practices.

This is a very strong paper that without a doubt is a clear acceptance. The communication is clear, reasoning is well-documented, and, although TMLR does not require it, there is immense novelty in the potential research directions this paper enables. The contributions of this research paper are incredibly interesting and useful to my own individual RL research and I enthusiastically recommend immediate acceptance. Myself, and I know many others, would love to incorporate this testbed into measuring our RL research.

**Additional Comments:**

Questions:
1. (Figure 1) Can you explain the reward function being computed based on a distance metric given a target value, and what that might necessarily mean?
2. The crystals found by generative models from Gruver et al. (2024) on page 2, it says M3GNet determines 50% are stable, whereas DFT determines only 11% are actually stable. To confirm, M3GNet and DFT agree on the same crystals, but M3GNet incorrectly labels others as stable when they are not? Did DFT find or determine any were stable that M3GNet could not?
3. How necessary is the modeling of this problem as a sequential decision-making task? I understand it is used to formulate it as a RL problem, but the assignments of each atom in the graph representing the crystal structure could perhaps be done in "one action" since the resulting "next state" is deterministic, right? For instance, if I put a atom Li in node 0, the next state would consist of this exact change - no non-determinism present. I suppose my curiosity on this came from the fact that the DFT is computed at the end of the episode, so there is no immediate reward present between state-action pairs. Is there a possible intermediate/immediate reward based on this approach, or would you say that is left for others in future RL research?
4. Why a batch size of 1 for RL fine-tuning on page 10? Is it due to computational cost of DFT? Or memory?

**Audience:**

Yes

**Audience Explanation:**

A resounding yes; this is a very clear and strong acceptance. I am extremely excited to see this published some day soon, and hope to see it published in TMLR. I do not hold this opinion lightly. This is one of the few papers I have reviewed in my career I am eager to follow-up on. I know many RL professionals who would be fascinated by these new challenging RL tasks and eager to tackle them to begin contributing to material discovery. I am confident that this would be a very popular article in TMLR, and that RL professionals can meaningfully contribute to this extremely important field of science.

**Broader Impact Concerns:**

Absolutely none to declare.

**Claims And Evidence:**

Yes

**Claims Explanation:**

Yes, the authors conduct comprehensive experiments showcasing multiple methodologies and are up-to-date on their literature.

**Requested Changes:**

If space permits after addressing other reviewers' requested changes, I would utilize that last little bit of space on page 11 by moving a small finding/result into the main body; select the most appropriate and important material that couldn't fit into the page lengths. Also, I think you are allotted 12 pages maximum for main body text so you can actually add another page of findings from the appendix. Again, I would suggest selecting the most important information, which to me might be the reward functions B.4, DFT settings B.5, GNN details B.6, MEGNet B.6.1 (if it makes sense to move, it looks like there is an important stipulation of CHGNet in B.7). Also, a bit strange for MEGNet to be considered a subsubsection when it is the only one in B.6. I would also include the novelty and uniqueness in the main body text where appropriate to fully utilize the 12 pages.

---

> ### Author Response · Authors · 2026-03-28
> **Author response to Reviewer zyuu (1/2)**
>
> We sincerely thank Reviewer zyuu for their thorough and enthusiastic review. We are delighted that the reviewer finds CrystalGym to be a valuable contribution to both the RL and materials science communities. In response to the reviewer's suggestions, we have expanded the main body of the paper to include additional content from the appendix (e.g. Novelty and uniqueness, and explicit references in the main text to sections and tables in the appendix). Modifications are mentioned in blue text in the current revised version.
> ___
> We address the reviewer's questions in detail below, with some additional analysis.
>
> ***Can you explain the reward function being computed based on a distance metric given a target value, and what that might necessarily mean?***
>
> The reward function is designed to encourage the agent to produce crystals whose DFT-computed property value p is close to a desired target value p̂. At the end of each episode (once all atomic positions are filled), the DFT simulator returns a scalar $p$, and the reward $r(s_N)$ is computed as a distance between $p$ and $p̂$. For bulk modulus, we use a scaled absolute distance $r(s_N) = \max(−|p − p̂|/p̂, −5)$, which is always non-positive and clipped to avoid excessively large penalties. For density and band gap, we use an exponential distance, which maps the deviation from the target to a value in $(0, 1]$, with a reward of −1 in the case of a DFT failure. This design ensures that the reward signal always reflects closeness to the target, rather than simply maximizing or minimizing the property value.
>
> ___
> ***Clarification on stability predicted by machine learning models compared to DFT***
>
> M3GNet labels a crystal as stable if it predicts a negative energy above convex hull, and DFT then serves as the ground truth to verify this. The discrepancy between the two arises because machine learning interatomic potentials (MLIPs) like M3GNet are trained to approximate the DFT energies, but this approximation could be imperfect, particularly for compositions and structures underrepresented in the training data or for structures near the stability boundary where even small prediction errors can flip the stability classification. The Matbench Discovery benchmark [1] assesses the performance across a wide range of ML models, including M3GNet.
>
> While the scope of our current study does not include optimization for energy or stability explicitly, we computed the stability of 1,080 generated structures using the recently proposed LeMat-GenBench evaluation pipeline [2], which relies on Orb-v3 [3], a state-of-the-art MLIP that is more recent and accurate than M3GNet. We found that none of the generated structures were stable (≤ 0 eV/atom energy above the convex hull) or metastable (≤ 0.1 eV/atom energy above the convex hull). This is consistent with our expectation, as we do not optimize for thermodynamic stability during RL training, and is discussed in detail in Appendix D.3 of the revised version. At present, we are unable to verify the true stability with DFT since we do not have a protocol to reliably estimate this quantity with Quantum Espresso.

---

> ### Author Response · Authors · 2026-03-28
> **Author response to Reviewer zyuu (2/2)**
>
> ***Necessity to formulate this as a sequential decision problem rather than a direct prediction of the entire composition in a single step? Is there a possible intermediate/immediate reward based on this approach, or would you say that is left for others in future RL research?***
>
> The transition function in CrystalGym is deterministic: placing atom $a_i$ at position $x_j$ results in a fully predictable next state. However, we frame the problem as a sequential decision-making task rather than predicting all the atoms at once, primarily because the latter would reduce the problem to a bandit setting with an action space of size $|A|^N$, where $|A|$ is the number of available elements and N is the number of atomic sites. Even for the smallest setting in our benchmark ($|A| = 18, N = 4$), this yields over 100,000 possible compositions, making direct one-shot search intractable. Besides, our framework is orthogonal to other crystal generation models (e.g. diffusion) that do not sequentially construct the crystal atom-wise. The sequential formulation allows the agent to build up a composition incrementally and reuse learned representations across partially complete states. Regarding intermediate rewards, in the current formulation, the reward is purely terminal: DFT is only called once per episode at the end. Designing effective intermediate reward signals is indeed a promising direction for future work.
> ___
> ***Why a batch size of 1 for RL fine-tuning on page 10? Is it due to computational cost of DFT? Or memory?***
>
> The batch size of 1 is primarily due to the computational requirements of DFT. Each episode requires a full DFT simulation to obtain the terminal reward, which takes several seconds per crystal depending on the property (see Table 4). Running multiple episodes in parallel to form larger batches would require the use of asynchronous DFT computation schemes across different GPUs, which were difficult to establish in this study. Since the goal of the LLM fine-tuning case study is to assess the feasibility of aligning LLM policies with DFT-based reward signals rather than to optimise training throughput, we adopted the simplest possible setup with gradient updates after each episode. Exploring larger batch sizes through effective parallelism of DFT computations is a natural next step.
>
> ___
> We thank the reviewer for the questions and suggestions, which has helped us improve the clarity of the paper. We are happy to answer any further questions or provide additional clarifications.
>
> **References**
>
> [1] Riebesell, Janosh, et al. "Matbench Discovery--A framework to evaluate machine learning crystal stability predictions." arXiv preprint arXiv:2308.14920 (2023).
>
> [2] Betala, Siddharth, et al. "LeMat-GenBench: A Unified Evaluation Framework for Crystal Generative Models." arXiv preprint arXiv:2512.04562 (2025).
>
> [3] Rhodes, Benjamin, et al. "Orb-v3: atomistic simulation at scale." arXiv preprint arXiv:2504.06231 (2025).

---

### Review · Reviewer_eMwi · 2025-12-19

**Summary Of Contributions:**

The paper presents CrystalGym, an RL benchmark that embeds DFT-in-the-loop evaluations as rewards, aiming to expose the real bottlenecks of applying reinforcement learning to materials discovery. The motivation of replacing surrogate rewards with higher-fidelity physical evaluations is well-founded. However, the current formulation raises concerns regarding practical utility, reproducibility, and contribution clarity.

## Strengths
- Compared to surrogate models, using DFT more effectively exposes the real bottlenecks encountered in real materials discovery workflows.
- The evaluation somehow reveals concrete failure modes of current RL methods under expensive and unreliable reward signals (e.g., poor sample efficiency, instability on band gap tasks).

## Weaknesses
- DFT evaluations are prohibitively expensive, making the proposed environment impractical for large-scale experimentation; this substantially limits reproducibility and hinders broad adoption by the community.
- The proposed framework largely amounts to a lightweight engineering wrapper that exposes DFT evaluations as a callable reward function, with limited novelty in terms of benchmark design or methodology.
- The proposed environment falls short of a standard benchmark, as it offers limited and narrow test cases and lacks a reproducible, well-aligned evaluation protocol that would enable consistent comparison across methods and research groups.
- The online RL formulation is weakly aligned with dominant materials discovery workflows, which typically rely on offline data and surrogate-assisted optimization under strict query budgets.
- Evaluation remains fully within the same DFT setup used for reward computation, leaving the external validity of the results (e.g., transfer to more realistic settings) unestablished.

**Audience:**

Yes

**Audience Explanation:**

It explores the use of reinforcement learning under expensive and unreliable reward signals, a setting that is relevant to many real-world scientific and engineering applications beyond materials discovery.

**Claims And Evidence:**

No

**Claims Explanation:**

A comparison with OpenAI Gym helps illustrate why the proposed environment does not provide sufficient evidence to support the claim of enabling sustained progress in reinforcement learning for materials discovery.
| Aspect | OpenAI Gym | Proposed Environment (CrystalGym) |
|------|------------|-----------------------------------|
| Method | Accelerate RL research by replacing real-world interactions with simulation | Embedding expensive physical evaluations |
| Evaluation cost | Cheap and fast simulation | Expensive and time-consuming DFT evaluations |
| Reproducibility | High: scalable, repeatable, and low-cost | Limited: high cost and failure rates hinder reproducibility |
| Standardized benchmarks | Large and diverse suite of well-established benchmarks | Limited and narrow set of standardized test cases |
| Role of environment | Abstraction that enables rapid algorithmic iteration | Thin interface exposing DFT as a reward signal |

**Requested Changes:**

- The paper would benefit from a clearer evaluation protocol that balances realism with practical usability. In particular, adopting a multi-fidelity (e.g., surrogate models → low-accuracy DFT → full DFT → optional higher-fidelity methods) could substantially reduce cost while preserving physical relevance.

- Given the expense of DFT-based rewards, reformulating the environment toward an offline (batch) RL or offline-to-online setting may better align with both reproducibility requirements and prevailing materials discovery workflows. Providing standardized offline datasets generated under fixed DFT budgets, together with a limited number of queries for final evaluation or policy refinement, would improve comparability and community adoption.

- Finally, the benchmark would be strengthened by including more test cases and strong non-RL baselines commonly used in materials discovery (e.g., Bayesian optimization, evolutionary search, or generative models with screening).

---

> ### Author Response · Authors · 2026-03-28
> **Author response to Reviewer eMwi (1/2)**
>
> We thank Reviewer eMwi for their careful reading and constructive feedback. We appreciate the reviewer's recognition that CrystalGym addresses a relevant and challenging problem. We address each of the raised concerns below, with some additional analysis. Modifications are mentioned in blue text in the current revised version.
> ___
>
> ***The proposed framework largely amounts to a lightweight engineering wrapper that exposes DFT evaluations as a callable reward function, with limited novelty***
>
> While the interface between RL and DFT may appear straightforward, to our knowledge, no prior work has proposed an open-source RL environment that directly incorporates DFT-based reward signals for property-driven crystal composition optimization. The environment provides a default setting required to make DFT calculations reasonably fast, reproducible, and consistent across a wide range of crystal compositions. Hence, we believe this reduces the domain barrier for RL researchers. Second, the benchmark design, including the choice of properties, reward formulations, action space subsets, and the single vs. mixed structure settings reflects decisions grounded in both RL and materials science. We hope the community will build on and expand CrystalGym framework for more challenging problems in materials discovery.
> ___
> ***Comparisons with dominant materials discovery workflows that use offline data and surrogate models.***
>
> CrystalGym is orthogonal to other dominant material discovery workflows that rely on offline datasets with surrogate models: rather than learning a distribution over known crystals, our RL agent explores the chemical space directly through interaction with DFT, without relying on any pre-existing labeled dataset of crystal properties. Labeled datasets for some properties are scarce: for instance, only ~7% of all crystals in the Materials Project have bulk modulus labels. For band gap, the distribution of existing crystals is highly skewed towards near-zero band gap (metallic behavior). Training accurate ML proxies for these properties is therefore an open and challenging problem. By learning directly from DFT signals, CrystalGym exposes the true difficulty of these properties and demonstrates how learnable high-fidelity reward signals are in principle. We therefore view online RL with DFT rewards not as a replacement of existing generative workflows, but as a complementary and necessary research direction: one that becomes increasingly important where data-driven approaches reach their limits.
> ___
> ***DFT evaluations are prohibitively expensive, making the proposed environment impractical for large-scale experimentation***
>
> We acknowledge that DFT computations are expensive. The primary goal of CrystalGym is to assess the complexity of the problem - *"How many steps does it take to reach a high reward with pure online RL?"*, and *"How learnable are DFT signals in general?"*, *“Which properties are easy to optimize for?”* which is exactly what our work focuses on. Real-world materials discovery workflows ultimately require high-fidelity atomic simulation-based evaluation before experimental verification, regardless of whether fast surrogate models are used earlier in the machine learning pipeline. Understanding how RL algorithms behave under such expensive and noisy reward signals is therefore crucial for scaling property-driven material generation to real-world settings. The absence of reliable labeled data and accurate surrogates for certain properties makes DFT the only viable source of reward signals. We also note that CrystalGym is modular by design, and the DFT calculator could be replaced with any other simulator, including faster ML potentials, making it straightforward for the community to experiment with reduced-cost variants.

---

> ### Author Response · Authors · 2026-03-28
> **Author response to Reviewer eMwi (2/2)**
>
> ***On narrow test cases and non-RL baselines.***
>
> We acknowledge that expanding the benchmark with more test cases and baselines would strengthen the work, and we plan to do so in future iterations. Property-conditioned crystal generation is an unsolved challenge, and RL is just one approach that can directly optimize for target property values without requiring labeled data. Including non-RL baselines (e.g. Bayesian optimization, evolutionary search, generative models, etc.) would require adapting them to the same DFT-in-the-loop setting, which is an interesting yet challenging research direction we hope CrystalGym will motivate. In Section 2 (Related Works) of the revised version, we have acknowledged evolutionary search as a useful direction for these problems.
> ___
> ***Multi-fidelity workflow / offline-to-online workflows***
>
> We conducted additional experiments exploring a model-based RL approach, where a learned reward model replaces DFT calls during RL policy training and is periodically fine-tuned with DFT-queried data: this setting aligns with the reviewer's suggestion of a surrogate-assisted protocol. We trained a reward model for band gap prediction using the Conservative Objective Model (COM) objective [1], adapting it to the discrete action space of atom types. This reward model was then used to train a DQN agent, with DFT queried once every 50 episodes to fine-tune the reward model online. The results demonstrates a fundamental challenge: the COM-based reward model collapses rapidly upon fine-tuning, and the true DFT-evaluated band gap values do not converge toward the target.
>
> When the reward model is used without any fine-tuning, the agent appears to learn according to the surrogate reward, but DFT evaluation of the resulting policy yields either near-zero band gaps or DFT failures. This confirms that the noise structure of DFT-based rewards is different from the settings where model-based RL and surrogate-assisted optimization have been successfully applied. The accuracy of physics-based property prediction models, particularly for challenging properties like band gap, remains an active area of research. We therefore maintain that directly learning from DFT signals, as proposed in CrystalGym, is an important first step before model-based or multi-fidelity approaches can be meaningfully evaluated. Appendix D.2 in the current version of the paper contains the results of these experiments.
> ___
> ***Reframing this to a batched (Offline) RL.***
>
> Regarding offline RL, generating standardized offline datasets of sufficient diversity and quality for property-driven crystal optimization would require a significant number of DFT calculations. Moreover, offline RL methods are known to struggle with out-of-distribution generalization, especially in this exponentially large chemical search space we consider. We would like to point the reviewer to the prior work by [2], which explicitly investigated offline reinforcement learning for crystal design and highlighted multiple challenges for band gap optimization. CrystalGym was designed precisely as a response to those findings: by providing an online RL environment with direct DFT-based rewards, we enable the community to investigate whether these limitations can be overcome without relying on offline datasets or surrogate models.
> ___
> We hope these clarifications address the reviewer's concerns, and we welcome any further questions or discussion.
>
> **References**
>
> [1] Trabucco, Brandon, et al. "Conservative objective models for effective offline model-based optimization." International Conference on Machine Learning. PMLR, 2021.
>
> [2] Govindarajan, Prashant, et al. "Learning conditional policies for crystal design using offline reinforcement learning." Digital Discovery 3.4 (2024): 769-785.

---

### Review · Reviewer_MMpP · 2026-03-06

**Summary Of Contributions:**

This paper presents CrystalGym, an online reinforcement learning (RL) framework that uses density functional theory (DFT) calculations as reward signals for crystal structure generation. The environment formulates the design task as a sequential decision process: starting from a sampled cubic lattice with fixed atomic coordinates, the agent assigns atomic species to each site. This formulation differs from traditional crystal structure prediction or fully de novo generation to simplify the design space and make the RL setup more tractable. Within this setting, the objective is to optimize target physical properties such as bulk modulus, density, and band gap.

The paper aims to bridge RL research and materials discovery by providing a standardized environment where RL algorithms can be developed and evaluated using DFT calculations as expensive reward signals. By standardizing the environment and predefining the DFT calculation pipeline, the framework is designed to allow RL researchers to focus on algorithmic development under a controlled yet computationally demanding reward setting.

However, the experimental analysis remains limited. The experimental setup does not address the advantages or distinct challenges of the proposed expensive-reward scenario. The evaluation is mostly restricted to reporting returns over training steps or episodes, with little investigation into algorithmic behavior, reward properties, or existing metrics in the materials discovery domain. As a result, it remains unclear whether the benchmark provides novel insights for either the RL research community or practical materials discovery.

**Additional Comments:**

None

**Audience:**

Yes

**Audience Explanation:**

This work lies at the intersection of RL and materials science, and is therefore likely to attract interest from researchers in both domains. In particular, given the growing attention to materials science problems within the broader ML community, the reviewer believes that at least a subset of TMLR’s audience would find the topic relevant.

**Broader Impact Concerns:**

There are no specific ethical implications or broader impact concerns associated with this work that require further addressing.

**Claims And Evidence:**

No

**Claims Explanation:**

The paper presents several claims that lack adequate empirical support or logical consistency.

First, a central motivation of the framework is to provide high-fidelity reward signals via DFT. However, the authors explicitly state that they do not perform structure relaxation for the generated crystals. Since unrelaxed structures can yield physical properties that deviate substantially from their ground-state values, this omission directly undermines the premise of obtaining highly accurate property estimates through DFT. The paper does not adequately address this problem or justify why unrelaxed DFT calculations remain a reliable reward signal in their setting.

Second, the comparison among multiple RL algorithms in Section 5.1 primarily relies on return-versus-step plots. While such plots indicate overall learning progress, they provide few insights into how the specific mechanisms of each algorithm lead to the observed results. A more granular analysis of the generated trajectories and data is necessary to justify the findings.

Third, at the end of Section 5.2, the authors draw a contrast between their DFT-based approach and optimization through machine learning (ML) property predictors. However, no experiments using ML surrogate models, e.g., ML interatomic potentials (MLIPs), are provided to support this claim.

**Requested Changes:**

### Critical
- Although the authors provide justifications for not reporting traditional generative metrics such as diversity, uniqueness, and novelty, the reviewer believes that these metrics should still be evaluated. Even if they are not the primary focus, reporting them would help reveal the true advantages and limitations of the proposed method.
- As discussed above, the comparison of RL algorithms in Section 5.1 requires deeper analysis beyond return curves to yield meaningful insights from the benchmark.
- The environment should support MLIP models as an alternative reward source. While the motivation for using DFT as an accurate but expensive reward is convincing, the current implementation does not include structure relaxation, limiting the accuracy that DFT is supposed to provide. In this case, MLIPs offer a practical alternative that can rapidly deliver consistent property evaluations within the benchmark, while also enabling researchers to isolate the effect of reward cost on learning.
- Authors may discuss how the proposed framework compares to evolutionary algorithm-based approaches for crystal structure optimization. Evolutionary methods are well-established baselines in this domain, and clarifying the relative benefits of an RL-based formulation over such methods would provide stronger motivation for the proposed framework.

### Recommended
- In Section 2, the citation format for “SymmCD” should be corrected.
- In section 2, the publication year for “Crystal LLM” is missing.
- Some references cite arXiv preprints for papers that have since been formally published. These should be updated to their official publication venues.
- The text in Figures 3, 4, and 5 is too small to read comfortably and should be enlarged.
- Some informal expressions (e.g., “new way”) should be replaced with more precise and formal language.

---

> ### Author Response · Authors · 2026-03-28
> **Author response to Reviewer MMpP (1/2)**
>
> We thank the reviewer for their thorough review and detailed feedback. We appreciate the reviewer's recognition that this work is relevant to both the RL and materials science communities. In the revised version, we corrected the citation formats for SymmCD and CrystalLLM, and updated arXiv preprint references to their official publication venues where applicable. Modifications are mentioned in blue text in the current revised version.  We address each of the raised concerns below, with some additional analysis.
> ___
> ***On the validity of unrelaxed DFT as a reward signal***
>
> We acknowledge that unrelaxed DFT calculations yield property values that may differ from their relaxed counterparts. However, we emphasize that this does not undermine the validity of our reward signal for the following reasons. The scope of this benchmark is limited to optimizing the properties of unrelaxed structures: from an RL perspective, the agent gets feedback for the structure it generates and not for the final relaxed structure. More generally, the goal of CrystalGym is not to serve as a tool for precisely optimizing properties of fully relaxed structures, but to expose the genuine challenges of learning from expensive, noisy, and sometimes unreliable reward signals in the context of materials discovery. That said, the current environment allows enabling the relaxation mode.  Structure relaxation requires multiple sequential DFT calculations per sample, making it computationally expensive in an online RL setting where thousands of episodes are required. We discuss this in Section 5.2 (Overall analysis) and explicitly in Appendix A, where we explain this design choice and identify the incorporation of ML potentials for faster relaxation as an immediate next step.
> ___
> ***On the depth of analysis of RL algorithms in Section 5.1***
>
> We agree that a more granular analysis of algorithmic behavior would be valuable. We note that CrystalGym is primarily an applied benchmark paper rather than a theory- or algorithm-driven work. From an applied RL perspective, the most natural and interpretable metrics are those that directly characterize learning progress and sample efficiency, which is what our return curves are designed to convey.
>
> That said, we agree that quantifying sample efficiency more rigorously would strengthen the analysis. To this end, we computed the area under the return curve (AUC) for each algorithm and property for one of the experiment (single crystal C2, easy targets), which provides a single scalar summary of both the speed of learning and the quality of the converged policy. The updated version of the paper includes this analysis in Appendix D.1.
>
>
> | x10^4 | BM | Dens. | BG |
> |---|---|---|---|
> | PPO | **14.7** | **5.9** | 11.3 |
> | Rainbow | 13.9 | 5.7 | **11.6** |
> | DQN | 13.8 | 5.7 | 10.5 |
> | SAC | 12.9 | 5.4 | 5.6 |
>
> These results are consistent with the qualitative observations in the paper: PPO achieves higher sample efficiency for bulk modulus and density, where the reward landscape is smoother and DFT failure rates are low, while Rainbow performs best for band gap, where its experience replay helps it learn from the rare high-reward states encountered during exploration. SAC consistently shows the lowest AUC across all properties, since it struggles to escape the exploration phase in this setting. We will include AUC analysis across all crystals and experiments in the revised version to provide a more systematic and quantitative characterization of algorithmic behavior.

---

> ### Author Response · Authors · 2026-03-28
> **Author response to Reviewer MMpP (2/2)**
>
> ***Using surrogate models (e.g. MLIPs) in the experiments, possibly as an alternate reward source.***
>
> CrystalGym is orthogonal to other material discovery workflows that rely on offline datasets with surrogate models, and is primarily aimed at learning directly from complex, non-differentiable reward signals like DFT. Labeled datasets for some properties are scarce: only ~7% of crystals in the Materials Project have bulk modulus labels, and band gap distributions are heavily skewed toward near-zero values, making accurate ML proxies an open challenge. MLIPs can generally be used for stability rewards and structure relaxation, but accurate computation of elastic properties remains difficult due to challenges in modeling the potential energy surface [1].
>
> We conducted additional experiments exploring a model-based RL approach, where a learned reward model replaces DFT calls during RL policy training and is periodically fine-tuned with DFT-queried data. We trained a reward model for band gap prediction using the Conservative Objective Model (COM) objective [2]. This reward model was then used to train a DQN agent, with DFT queried once every 50 episodes to fine-tune the reward model online. The results demonstrate that COM-based reward model collapses rapidly upon fine-tuning, and the true DFT-evaluated band gap values do not converge toward the target.
>
> When the reward model is used without any fine-tuning, the agent appears to learn according to the surrogate reward, but DFT evaluation of the resulting policy yields either near-zero band gaps or DFT failures. This confirms that the noise structure of DFT-based rewards is different from the settings where model-based RL and surrogate-assisted optimization have been successfully applied. We therefore maintain that directly learning from DFT signals, as proposed in CrystalGym, is an important first step. Appendix D.2 in the current version of the paper contains the results of these experiments.
>
> ___
> ***Diversity, uniqueness, and novelty metrics***
>
> In Section 5.2 of the revised version, we report that over 60% of the ~3,000 compositions generated by all trained policies across all experiments were novel (using an exact composition match criteria) with respect to the Materials Project, and that the average fraction of unique compositions per experiment was 48.5%. Additionally, we performed a small-scale evaluation using the LeMat-GenBench protocol [3] on around 1080 unique structures. Appendix D.3 and Table 6 in the revised version show this analysis. Given that CrystalGym is not a generative model, these metrics are reported for completeness rather than as primary evaluation criteria. Out of the 1,080 structures, 63.6% were valid and 89.2% were novel with respect to the LeMat-Bulk reference dataset. Diversity scores were lower (25.4% element coverage, 10.0% space group coverage), which is expected given our restricted action spaces and limited seed structure variability.
> Regarding stability, none of the structures were stable (≤ 0 eV/atom above the convex hull) or metastable (≤ 0.1 eV/atom), with a mean energy above hull of 1.835 eV/atom estimated by ORBv3. This is expected since we do not explicitly optimize for thermodynamic stability during RL training. We note that stability rates below 5% are common even for dedicated generative models on LeMat-GenBench, and we propose multi-objective rewards jointly targeting the desired property and thermodynamic stability as a promising future direction for CrystalGym.
> ___
> ***Comparison with evolutionary algorithms***
>
> We acknowledge that traditional evolutionary approaches for crystals, like USPEX, may potentially allow optimization of composition given a fixed structure with a customizable reward scheme. We agree that discussing them in the context of CrystalGym would strengthen the motivation for the RL formulation, and have added some discussion in Section 2 (Related Works). While we are unable to perform a comprehensive set of experiments with traditional evolutionary approaches given the tight time constraints, we have included some relevant citations in the revised version of the submission. We consider evolutionary baselines as immediate future work.
> ___
> We hope these clarifications adequately address the reviewer's concerns. We are also working on improving the visibility of the text in all the plots, and will update the final version accordingly. We welcome any further questions or discussion.
>
>
> **References**
>
> [1] Gao, Pengfei, and Haidi Wang. "Benchmarking Universal Machine Learning Interatomic Potentials for Elastic Property Prediction." arXiv preprint arXiv:2510.22999 (2025).
>
> [2] Trabucco, Brandon, et al. "Conservative objective models for effective offline model-based optimization." International Conference on Machine Learning. PMLR, 2021.
>
> [3] Betala, Siddharth, et al. "LeMat-GenBench: A Unified Evaluation Framework for Crystal Generative Models." arXiv preprint arXiv:2512.04562 (2025).

---

### Decision · Action_Editor_8o3u · 2026-04-21

**Recommendation:** Reject

**Additional Comments:**

The authors should rework and pay close attention to MMpP's comment and commits to a clear direction and strengthen the experimental design.

**Audience:**

Yes

**Audience Explanation:**

Yes, if successful RL and material science research community would have been interested.

**Claims And Evidence:**

No

**Claims Explanation:**

The paper introduces CrystalGym, an open-source reinforcement learning environment that utilizes Density Functional Theory (DFT) calculations as direct reward signals to optimize the composition of crystalline materials. The reviewers appreciated that the paper  introduces a real-world application domain for RL and its ability to communicate complex materials science concepts clearly to a non-expert audience. It was also recognized as a potentially vital testbed for studying how RL algorithms handle expensive and noisy reward signals. However, the reviewers found that the paper suffers from a fundamental lack of direction failing to provide meaningful value to either RL research or materials discovery.

To address reviewer concerns, the authors conducted additional surrogate model experiments, incorporated quantitative sample efficiency and standardized generative metrics, expanded the main text with previously appended technical details, and provided detailed justifications for their design choices regarding unrelaxed DFT reward signals and the sequential MDP formulation

However, even though reviewers recognized the potential of this interdisciplinary topic, the concerns around strengthening experimental design remain and the paper falls short of its claims to advance both RL research and practical materials discovery. To be considered for the publication, the authors need to make the benchmark meaningful and useful for the material science.

**Resubmission Of Major Revision:**

The authors may consider submitting a major revision at a later time.